# A bacterial immunomodulatory protein with lipocalin-like domains facilitates host–bacteria mutualism in larval zebrafish

Annah S Rolig[1], Emily Goers Sweeney[1], Lila E Kaye[1], Michael D DeSantis[1], Arden Perkins[1], Allison V Banse[1], M Kristina Hamilton[2], Karen Guillemin[1,3]*

[1]Institute of Molecular Biology, University of Oregon, Eugene, United States; [2]Institute of Neuroscience, University of Oregon, Eugene, United States; [3]Humans and the Microbiome Program, Canadian Institute for Advanced Research, Toronto, Canada

**Abstract** Stable mutualism between a host and its resident bacteria requires a moderated immune response to control bacterial population size without eliciting excessive inflammation that could harm both partners. Little is known about the specific molecular mechanisms utilized by bacterial mutualists to temper their hosts' responses and protect themselves from aggressive immune attack. Using a gnotobiotic larval zebrafish model, we identified an *Aeromonas* secreted immunomodulatory protein, AimA. AimA is required during colonization to prevent intestinal inflammation that simultaneously compromises both bacterial and host survival. Administration of exogenous AimA prevents excessive intestinal neutrophil accumulation and protects against septic shock in models of both bacterially and chemically induced intestinal inflammation. We determined the molecular structure of AimA, which revealed two related calycin-like domains with structural similarity to the mammalian immune modulatory protein, lipocalin-2. As a secreted bacterial protein required by both partners for optimal fitness, AimA is an exemplar bacterial mutualism factor.
DOI: https://doi.org/10.7554/eLife.37172.001

*For correspondence: kguillem@uoregon.edu

## Introduction

The vertebrate gastrointestinal tract harbors the highest density of microbes recorded for any microbial habitat (*Whitman et al., 1998*). These microbial communities are not only tolerated by their hosts, but are also required for normal host development and physiology, highlighting the intimacy of these mutualisms (*McFall-Ngai et al., 2013*). How these host-microbe mutualisms are established and maintained is unresolved. Understanding these processes will require identifying the molecular mechanisms by which resident microbes modulate the host immune system and defining the benefit these microbes derive from investing in such interactions. This knowledge is critical to dissect the microbial etiology of a range of diseases associated with excessive inflammation, including inflammatory bowel disease (IBD), diabetes, metabolic syndrome, and sepsis (*Huttenhower et al., 2014*; *Turnbaugh et al., 2006*; *Wen et al., 2008*). Furthermore, a thorough understanding of these host-microbe mutualisms will accelerate the development of effective treatments for these inflammatory diseases. Here, we explore the host-bacterial mutualism between zebrafish and their prominent colonizer, *Aeromonas,* and identify a secreted bacterial protein that promotes both bacterial colonization and host survival by preventing excessive inflammation.

Pathogenic bacterial species are often defined by unique virulence factors that enable host colonization and determine the severity of host disease symptoms. Much less well known are the unique

**eLife digest** Animals, including humans, harbor vast numbers of bacteria inside our digestive tracts. But rather than wage constant war, we have learned to coexist peacefully, and many of these bacteria are important to keep us healthy. Our immune system controls the number of bacteria, but in some diseases, this balance fails, and immune cells called neutrophils start a defense response. However, such attacks also cause inflammation in our guts, which can damage the tissues and organs.

Understanding the delicate balance between immune cells and individual bacteria in humans remains a challenge. Our gut bacteria live in complex communities with many species of microorganisms. Using animals like zebrafish can help to find out if gut bacteria are able to prevent such inflammation. These fish can grow under sterile conditions in the laboratory, allowing the study of added individual bacteria and the molecules they release. They are also transparent, making it easy to capture images of their organs under the microscope. In the wild, they live side-by-side with a type of bacteria called *Aeromonas*.

Here, Rolig et al. added *Aeromonas* bacteria to sterile zebrafish and observed how they interacted with the fish immune system. This revealed that the bacteria produce a protein to pacify immune cells, named 'Aeromonas immune modulator' (AimA). The sequence of this new protein did not look like any known molecules, but its 3D structure resembled a protein found in animals called lipocalin. AimA dampened inflammation in the gut.

When *Aeromonas* without AimA colonized the zebrafish, their neutrophils went to war. Their guts became inflamed and the bacteria started to die. But when the zebrafish had no neutrophils in their gut, nothing could stop the AimA-deficient bacteria and they grew too much. They produced toxic products, triggering septic shock and killing the fish. Adding AimA back into the fish stopped the inflammation and prevented septic shock, restoring balance. Both partners need AimA to survive. The bacteria need it to shield them from the immune response and the fish need it for protection against toxic products made by the bacteria.

Understanding proteins like AimA could help us to control bacteria and gut inflammation. Further work could reveal ways to use similar molecules to treat inflammation or to boost the growth of friendly bacteria.

DOI: https://doi.org/10.7554/eLife.37172.002

factors produced by anti-inflammatory bacterial species, which likely play a key role in establishing host-bacterial mutualism and maintaining intestinal homeostasis. One such factor is the zwitterionic polysaccharide, PSA, produced by the human symbiont *Bacteroidetes fragilis,* that induces a toleragenic T cell profile and protects against intestinal inflammation (*Mazmanian et al., 2008*; *Mazmanian et al., 2005*; *Round and Mazmanian, 2010*). The anti-inflammatory properties of *Lactobacillus rhamnosus* have been attributed to secreted proteins p75 and p40 (*Yan et al., 2007*), and *Faecalibacterium prausnitzii* secretes an anti-inflammatory 15 kDa protein called microbial anti-inflammatory molecule or MAM (*Yan et al., 2007*; *Quévrain et al., 2016*; *Sokol et al., 2008*; *Carlsson et al., 2013*; *Martín et al., 2014*). These secreted proteins reduce inflammation in mouse models of colitis (*Yan et al., 2011*), which suggests their therapeutic potential, but we do not yet understand how these proteins aid the bacteria that produce them. Investigating the bacterial fitness benefits for producing anti-inflammatory activities is critical for developing approaches to promote membership of these immunoregulatory bacterial species in chronic inflammatory diseases like IBD and to use bacterial immune modulators as therapeutics to shape microbiota composition.

The investigation of microbiota-derived immunomodulatory factors is challenging because their effects on the host are subtler then those of disease-causing toxins and they are produced by complex and genetically intractable microbial consortia. We used the zebrafish, *Danio rerio,* as a high-throughput, gnotobiotic model vertebrate system to identify individual bacterial products with immunomodulatory properties and understand how those products promote host-bacterial mutualism. Zebrafish have long been used as a vertebrate model for developmental biology because of their optical transparency, high fecundity, and rapid early development (*Grunwald and Eisen, 2002*). Zebrafish embryos develop ex-utero within their protective chorions and first encounter

environmental microbes when they hatch as larvae between 2 and 3 days post fertilization (dpf). By 5 dpf the animals have a fully functional digestive tract that is colonized with bacteria (*Bates et al., 2006*; *Stephens et al., 2016*; *Wallace et al., 2005*). The larvae are also equipped with a functional innate immune system with highly dynamic neutrophils and macrophages that serve as the primary cellular defenses before the emergence of adaptive immune cells at approximately 3 weeks of age (*Renshaw and Trede, 2012*). The zebrafish is an excellent model for studying the establishment of host-bacterial symbioses (*Burns and Guillemin, 2017*) because hundreds of zebrafish can be easily derived and maintained in a gnotobiotic state with defined microbial isolates during larval development (*Melancon et al., 2017*). The large population sizes of animals allow for the analysis of the subtle effects of resident microbes on host phenotypes. For example, we have shown that intestinal neutrophil populations in larval zebrafish range from an average of 3 cells in germ free (GF) animals to 7 cells in conventionally reared (CV) animals, a difference that can be readily detected in well-powered experiments despite extensive inter-host variation (*Rolig et al., 2015*). Our use of gnotobiology to assay the effect of individual microbes and their products on the host has been aided by the fact that many zebrafish bacterial isolates are culturable, have known genome sequences (*Stephens et al., 2016*), and are genetically tractable (*Hill et al., 2016*; *Wiles et al., 2016*). This gnotobiotic approach, however, fails to capture the complexity of naturally occurring vertebrate microbiota. Additionally, in the zebrafish we are largely limited to using gnotobiology to study the larval period because we have yet to develop adequate nutritional and husbandry approaches to promote the growth of GF zebrafish to adulthood.

In this study we focused on the mutualism established between larval zebrafish and their resident *Aeromonas* species. *Aeromonas* is an important symbiont of the zebrafish and the sole bacterial genus that we found present in 100% of individuals at all developmental time points in a comprehensive longitudinal study of zebrafish gut microbiomes across their lifespan (*Stephens et al., 2016*). Furthermore, we have shown that in mono-associations, representatives of the *Aeromonas* genus can fully rescue defects in glycan expression (*Bates et al., 2006*), epithelial cell proliferation (*Cheesman et al., 2011*), innate immune response (*Rolig et al., 2015*), and pancreatic beta cell expansion (*Hill et al., 2016*) that are characteristic phenotypes of GF fish. We have examined the dynamics of the *Aeromonas*-zebrafish mutualism, taking advantage of the optical transparency of the larvae to monitor both bacterial and host cells in vivo (*Taormina et al., 2012*; *Jemielita et al., 2014*). Here we specifically quantified the innate immune response against *Aeromonas* bacterial genetic variants, using GFP-expressing intestinal neutrophils as a metric of the host response (*Renshaw et al., 2006*). Neutrophils are a primary component of the initial inflammatory response and are responsive to both infecting pathogens and resident intestinal microbes (*Harvie and Huttenlocher, 2015*; *Bates et al., 2007*; *Kanther et al., 2014*). This responsiveness makes the intestinal neutrophil population an ideal readout of the host-microbe mutualism between zebrafish and *Aeromonas*.

Using the gnotobiotic zebrafish model, we identified a potent immune modulatory protein secreted by *Aeromonas*, which we named Aeromonas immune modulator A (AimA) because it reduces intestinal inflammation. To understand the molecular mechanism of AimA's function, we determined the protein crystal structure, which revealed two related calycin-like domains that are structurally similar to the immune modulatory factor found in mammals, lipocalin-2 (LCN2). Consistent with AimA targeting the host immune response, as opposed to counteracting a specific bacterial product, we showed that purified AimA, and each of its subdomains, prevents chemically induced intestinal inflammation. Furthermore, we found that this activity is blocked by exogenous mouse LCN2 (mLCN2), implying a shared mechanism of action between these structurally similar bacterial and mammalian proteins. However, AimA does not appear to function like LCN2 as a binding protein for bacterial siderophores. To understand AimA's function for *Aeromonas*, we generated mutants lacking *aimA* and its homologue *aimB*. In mono-associations with *aim* mutants, we found that both the host and bacterial partners suffered reduced viability, which could be rescued with exogenous AimA protein. We showed that the fitness cost to the bacteria was due to excessive inflammation because the *aim* mutant survival deficit was erased in an immunocompromised host. In parallel, we showed AimA's benefit to the host extended to protection from lipopolysaccharide (LPS) intoxication. Together, these data provide an exemplar of a bacteria immune modulatory factor that is beneficial for both partners in a bacterial mutualism.

## Results

### Aeromonas secretes a protein that modulates the intestinal neutrophil response

Based on the mutualism between *Aeromonas* and zebrafish (*Stephens et al., 2016*), we hypothesized that *Aeromonas* would use specific secreted factors to co-exist with its host. We tested this hypothesis by comparing the zebrafish intestinal neutrophil response to mono-association with an *Aeromonas* isolate, *A. veronii* strain Hm21 (*Maltz and Graf, 2011*), versus mono-association with an isogenic mutant lacking the type two secretion system (T2SS)(ΔT2)(*Table 1*), a major system for secretion in Gram-negative bacteria (*Maltz and Graf, 2011*). We colonized each bacterial strain in zebrafish from 4 dpf to 6 dpf and measured the number of intestinal neutrophils that accumulated in response to each colonizing bacterial strain. In these experiments we compared the ΔT2 strain to a genetically complemented strain, ΔT2C, previously shown to restore T2SS function (*Maltz and Graf, 2011*), which allowed us to rule out the possibility that phenotypes associated with the ΔT2 strain were due to polar or second site mutations. The ΔT2C and ΔT2 colonized the zebrafish intestine to similar levels in mono-associations (*Figure 1—figure supplement 1*), however the ΔT2 mutant induced a greater intestinal neutrophil response than the ΔT2C (*Figure 1A*). This result suggests that *A. veronii* strain Hm21 secretes a product that decreases the neutrophil response to its colonization. The increased neutrophil response to ΔT2 was rescued by concurrent treatment with cell-free supernatant (CFS) collected from the ΔT2C strain, but not CFS collected from ΔT2 (*Figure 1A*). To identify the factor produced by ΔT2C and not from the ΔT2 strain that resulted in fewer intestinal neutrophils, we performed mass-spectrophotometry on the CFS from these two strains and determined which proteins were enriched in the ΔT2C compared to the ΔT2 strain. This analysis resulted in a list of 22 proteins that were enriched by greater than 10 counts in the ΔT2C CFS (*Supplemental file 1*). We narrowed down the potential candidates further by performing ammonium sulfate fractionation on the ΔT2C CFS and adding fractions to fish mono-associated with ΔT2. We found neutrophil modulating activity primarily in the 40–60% salt fraction, but also saw some activity in the 20–40% salt fraction (*Figure 1A*). An SDS PAGE gel of these fractions revealed a similar banding pattern for each fraction, except the bands in the 20–40% fraction were considerably weaker. We observed two prominent bands at 33 kDa and 55 kDa. These sizes corresponded to two proteins on our list of proteins enriched in the ΔT2C CFS, an uncharacterized protein (UP) and a chitin binding protein (CBP), respectively (*Supplemental file 1*). To test whether either of these proteins modulated the neutrophil response to ΔT2 colonization, we cloned the sequence for each gene into an overexpression vector and induced expression in *Escherichia coli*, resulting in *E. coli* CFS that was heavily dominated by the protein of interest (*Figure 1B*). We treated zebrafish mono-associated with ΔT2 with the CFS resulting from overexpression of each protein of interest and found neutrophil modulating activity in the *E. coli* CFS with the 33 kDa UP and not with CBP or control CFS from *E. coli* containing an empty expression vector (*Figure 1C*). We named this neutrophil-modulating protein *Aeromonas* immune modulator A (AimA).

**Table 1.** Strain table.

| Strain | Characteristics | Ref. or source | Manuscript abbreviation |
|---|---|---|---|
| Hm21S | Parent strain, Sm$^R$ | *Graf, 1999* | Aer |
| HE-1095 | Hm21S::interrupted *exeM* mTn5 Km$^R$ Sm$^R$ | *Maltz and Graf, 2011* | Aer ΔT2 |
| HEC-1344 | HE-1095::Tn7 containing Tp$^R$ *exeMN* + promoter region | *Maltz and Graf, 2011* | Aer ΔT2C |
| ASRC7 | Hm21S *aimA::cm$^R$* | This study | Aer ΔaimA |
| ASRD5 | Hm21S ΔaimB | This study | Aer ΔaimB |
| ASRD4 | Hm21S *aimA::cm$^R$*; ΔaimB | This study | Aer ΔAΔB |
| ZOR0001 | Zebrafish *Aeromonas* isolate | Stephens | ZF Aer |
| ASRC9 | ZOR0001 *aimA::cmR* | This study | ZF Aer ΔaimA |

DOI: https://doi.org/10.7554/eLife.37172.007

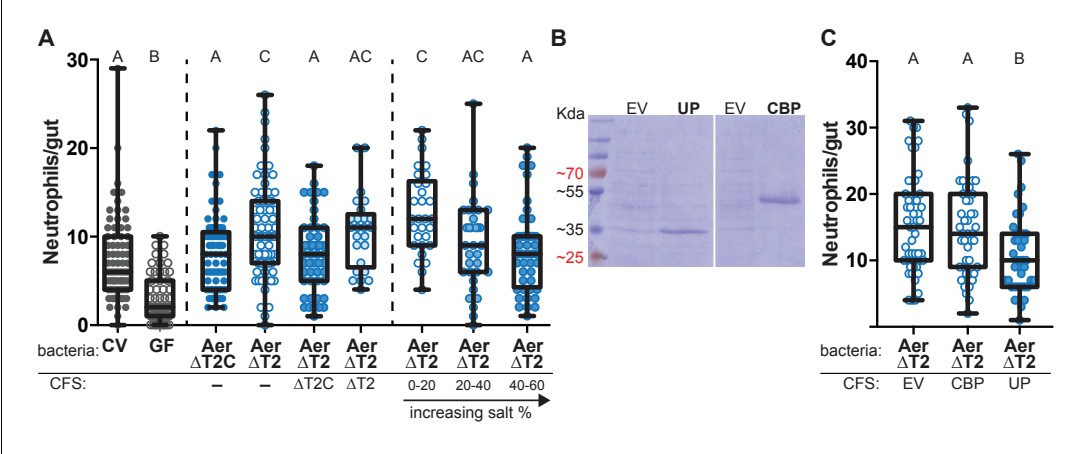

**Figure 1.** *Aeromonas* secretes a protein that regulates intestinal neutrophil response to a general model of inflammation. (**A**) Intestinal neutrophil response to conventional (CV), germ free (GF), and mono-associations of wild-type *Aeromonas* (Aer ΔT2C), and an isogenic mutant of the type II secretion system (ΔT2). ΔT2 induces a greater neutrophil response that is rescued by addition of cell free supernatant (CFS) from the wild-type *Aeromonas* strain. Ammonium sulfate fractionation narrowed potential candidates down to two proteins, an unidentified protein (UP) or chitin binding protein (CBP). (**B**) *E. coli* BL21 carrying pET21b expression vector overexpression of the candidate proteins of interest. (**C**) Addition of CFS containing UP rescued the increased neutrophil response induced by ΔT2. EV: empty vector control CFS. Letters denote p<0.05, ANOVA followed by Tukey's post hoc test. Each box represents the first to third quartiles, center bar the median, and whiskers the maximum and minimum. Each dot represents one fish; data collected from at least two independent experiments; n ≥ 24.

DOI: https://doi.org/10.7554/eLife.37172.003

The following source data and figure supplements are available for figure 1:

**Source data 1.** *Aeromonas* secretes a protein that regulates intestinal neutrophil response to a general model of inflammation.

DOI: https://doi.org/10.7554/eLife.37172.005

**Figure supplement 1.** Colonization of *A. veronii* strain Hm21 type two secretion mutant.

DOI: https://doi.org/10.7554/eLife.37172.004

**Figure supplement 1—source data 1.** Colonization of *A. veronii* strain Hm21 type 2 secretion mutant.

DOI: https://doi.org/10.7554/eLife.37172.006

## The AimA protein has two distinct calycin domains with related folds

DNA and amino acid homology searches revealed that no homologues to AimA exist outside of the *Aeromonas* genera. Further, analysis of the amino acid sequence of AimA with the N terminal amino acid secretion signal removed revealed a lack of sequence identity with known domains or structurally characterized proteins and therefore offered little insight into the structure or function of AimA. Thus, to gain insight into the mechanism of AimA, we constructed a His-tagged version that was purified from *E. coli,* and we crystallized the protein. We determined the molecular structure of AimA using a heavy atom derivative and Single-wavelength Anomalous Dispersion (SAD) phasing to a resolution of 2.3 Å (PDB 6B7L; *Table 2*).

The structure of AimA revealed two domains connected by a short linker (*Figure 2A,B*). β-strands dominate each domain with the carboxy terminal (C-term) domain forming a complete β-barrel and the amino terminal (N-term) domain containing a curved β-sheet. Structural homology searches of full length AimA against all Protein Data Bank-deposited structures using PDBeFold resulted in structures that aligned to only one domain or the other, with the majority aligning with the C-term domain (*Krissinel et al., 2004*; *Berman et al., 2000*). Therefore, we performed structural homology searches against each domain separately, which revealed that both domains had similarity to proteins in the calycin superfamily (*Supplemental file 2* and *Figure 2—figure supplement 1*).

The calycin superfamily of proteins contains members from all domains of life and includes lipocalins, fatty acid binding proteins, and avidins. This superfamily is defined by an anti-parallel β-barrel with a repeated +1 topology (*Flower, 1996*). Notably, the calycin superfamily is known for structural conservation without high amino acid sequence conservation (*Flower, 1996*; *Lakshmi et al., 2015*), consistent with a lack of sequence homology between AimA and other calycin family members. The N-term domain of AimA has an incomplete β-barrel, but maintains some structural homology to

**Table 2.** Data Collection and Refinement Statistics for Deposited Model.

| Data collection | Iodide | | | Native | | |
|---|---|---|---|---|---|---|
| space group | P622 | | | P622 | | |
| unit cell a, b, c (Å) | 161.4 | 161.4 | 66.6 | 160.5 | 160.5 | 66.2 |
| alpha, beta, gamma (degrees) | 90 | 90 | 120 | 90 | 90 | 120 |
| resolution (Å) | 50.0–2.7 (2.75–2.70) | | | 41.1–2.30 | | |
| completeness (%) | 100 (100) | | | 100 (100) | | |
| no. unique reflections | 14772 (711) | | | 22818 (3244) | | |
| multiplicity | 37.7 (38.1) | | | 70.2 (71.6) | | |
| I/sigma> | 44.2 (3.5) | | | 16.1 (1.4) | | |
| CC1/2 | 1 | 0.861 | | 1.0 (0.9) | | |
| CC1/2 anomolous | 1 | 0.873 | | | | |
| R merge (%) | 19.6 (486) | | | 19.1 (574) | | |
| Refinement | | | | | | |
| R work (%) | | | | 17.2 | | |
| R free (%) | | | | 20.4 | | |
| no. of molecules in the asymetric unit | | | | 1 | | |
| no. protein residues | | | | 290 | | |
| no. of waters | | | | 69 | | |
| rmsd for lengths (Å) | | | | 0.008 | | |
| rmsd for angles (deg) | | | | 1.1 | | |
| Ramachandran plot (%) | | | | | | |
| preferred | | | | 96.2 | | |
| allowed | | | | 3.4 | | |
| outliers | | | | 0.4 | | |
| Avg. B factor (Å2) | | | | | | |
| mainchain[*] | | | | 81 | | |
| waters | | | | 75 | | |
| new PDB entry | | | | 6B7L | | |

[*] Several loop regions display high mobility but have been modeled due to a visible chain path in the electron density, resulting in an increase in the average observed B-factors of the main chain

DOI: https://doi.org/10.7554/eLife.37172.008

streptavidin. Streptavidin binds biotin tightly, but we found no evidence that AimA binds biotin (*Figure 2—figure supplement 1*). The C-term domain of AimA has structural homology both to avidins and lipocalins (*Supplemental file 2*).

## In a model of intestinal inflammation, AimA reduces neutrophil influx

AimA's structural similarity to lipocalin proteins was intriguing because of these proteins' known roles in modulating neutrophil behavior (*Moschen et al., 2017*). Based on this structural similarity, we wondered whether AimA's influence on intestinal neutrophil numbers was specific to *Aeromonas* colonization or whether it acts as a more general immunoregulatory molecule to influence neutrophil behavior. To test whether AimA modulates neutrophil numbers in response to stimuli other than *Aeromonas* mono-association, we employed the zebrafish model of soysaponin-induced inflammation. Farmed fish, such as salmon and carp, fed soybean meal as a protein source are well known to develop intestinal inflammation, and zebrafish are a good model of this irritation (*Hedrera et al., 2013*). We fed conventionally raised zebrafish larvae Zeiglers fish food with 0.3% soysaponin from 4 dpf to 6 dpf and saw a significant increase in the number of intestinal neutrophils in response to the soysaponin, as expected (*Figure 3A*). When we treated the fish with 100 ng/mL purified AimA

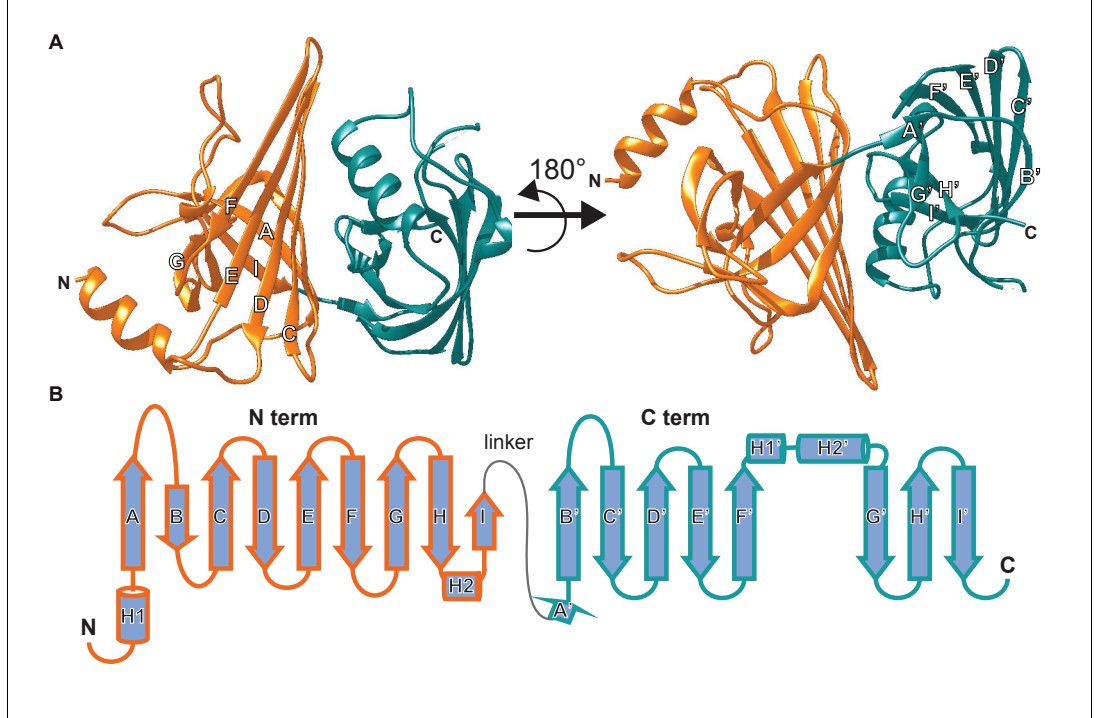

**Figure 2.** The crystal structure of AimA reveals two calycin domains. (**A**) The 2.3 Å structure of AimA displays two calycin domains (PDB ID 6B7L). (**B**) The amino terminal domain (orange) of AimA is connected by a short linker to the carboxy terminal domain (teal) and both contain an eight-stranded full (C-term) or partial (N-term) β-barrel.

DOI: https://doi.org/10.7554/eLife.37172.009

The following figure supplements are available for figure 2:

**Figure supplement 1.** Structural homology of the two domains of AimA to avidins.
DOI: https://doi.org/10.7554/eLife.37172.010
**Figure supplement 2.** A paired refinement demonstrates that the extended resolution improves the quality of the model.
DOI: https://doi.org/10.7554/eLife.37172.011

concurrently with soysaponin, AimA prevented the increase in neutrophil influx in response to soysaponin (*Figure 3A*). This result demonstrates that AimA alone can inhibit pro-inflammatory signaling pathways elicited by stimuli other than *Aeromonas*.

## Presence of mouse lipocalin-2 inhibits AimA function

Because AimA reduces neutrophil influx in a model of intestinal inflammation and the C-term domain has structural homology to lipocalin proteins (*Figure 3A*, *Supplementary file 2*), we hypothesized that the neutrophil reducing capacity of AimA was related to its lipocalin-like structure. Lipocalins are defined by three key structurally conserved regions, SCR1, SCR2, and SCR3; of these, SCR1 contains four key residues and is the most conserved (*Figure 3B*) (*Flower, 1996*; *Lakshmi et al., 2015*). The SCR1 in the C-term domain of AimA overlaps structurally with the SCR1 of mouse lipocalin-2 (mLCN2) and shares some related amino acids (*Figure 3B*). Given this structural similarity, we hypothesized that AimA may function in the same pathways as mLCN2. Thus, we asked whether the presence of mLCN2 protein would alter the capacity of AimA to reduce the neutrophil response to soysaponin. We found that the addition of mLCN2 concurrently with soysaponin did not influence the neutrophil response to soysaponin (*Figure 3C*); however, addition of mLCN2 in conjunction with AimA blocked the protective effect of AimA against soysaponin-induced inflammation. This result suggests that mLCN2 may interfere with AimA's activity by direct binding, competing for the same host receptor, or acting in a competing pathway.

LCN2's immunomodulatory activity is mediated, at least in part, by binding iron-sequestering siderophores, including enterobactin from *E. coli* (*Xiao et al., 2017*), thus we explored whether

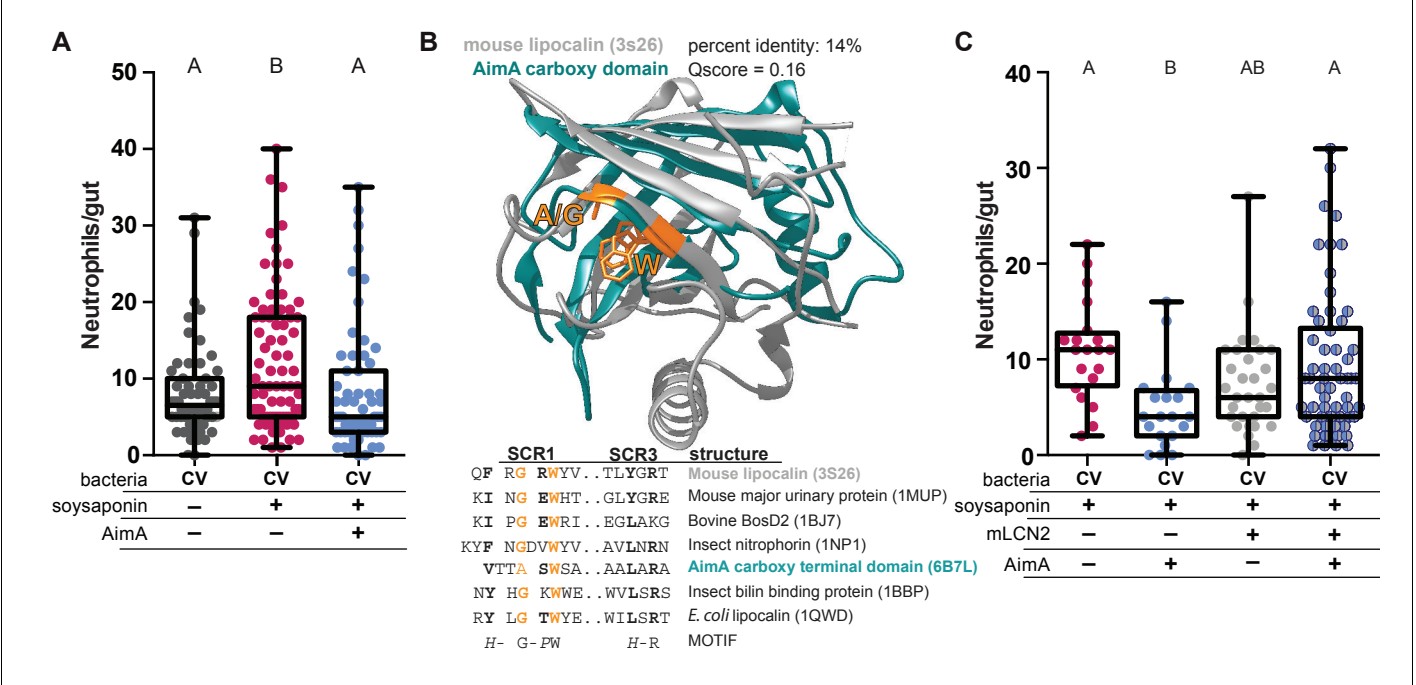

**Figure 3.** AimA contains regions structurally conserved with the lipocalin family and reduces soysaponin induced inflammation. (**A**) Feeding zebrafish soy saponin induces increased intestinal neutrophil response, and treating those fish with 100 ng/mL purified AimA prevents the increased intestinal neutrophil response. Letters indicate significantly different groups; ANOVA with multiple comparisons. Each dot represents one fish; data collected from at least two independent experiments; n ≥ 24. (**B**) Structural overlay of mouse lipocalin (PDB ID 3S26) and AimA C-term domain using PDBeFold. Qscore is a structural overlay quality score that takes into account both the root mean standard deviation (RMSD) of the $C_\alpha$ carbons and the alignment length. Qscore of 1 is perfect alignment, 0 is no alignment. The residues highlighted in orange are conserved in both the sequences and the structures (see table below). Displayed are the sequences of Structurally Conserved Regions (SCR) 1 and 3 of a representative set of kernel and outlier lipocalins, with PDB IDs in parenthesis. The C-term domain of AimA is included and contains a subset of conserved SCR residues. Bold residues are conserved across the sequences. Orange residues are conserved across the sequences and in the structures of mLCN and the C-term domain of AimA. (**C**) Treatment of conventionally raised (CV) fish with soysaponin and lipocalin prevents AimA from reducing the neutrophil response. Each dot represents one fish; n ≥ 20 from at least three independent experiments. Letters indicate significantly different groups; ANOVA with multiple comparisons.
DOI: https://doi.org/10.7554/eLife.37172.012

The following source data and figure supplements are available for figure 3:

**Source data 1.** AimA contains regions structurally conserved with the lipocalin family and reduces soysaponin induced inflammation.
DOI: https://doi.org/10.7554/eLife.37172.014

**Figure supplement 1.** AimA does not bind enterobactin.
DOI: https://doi.org/10.7554/eLife.37172.013

**Figure supplement 1—source data 1.** AimA does not bind enterobactin.
DOI: https://doi.org/10.7554/eLife.37172.015

AimA had a similar enterobactin-binding activity. The original evidence for LCN2's enterobactin binding came from the protein crystal structure of human LCN2, in which enterobactin was co-crystalized (*Goetz et al., 2002*). We did not find enterobactin in the AimA structure, which was expected because the *E. coli* strain, BL21 DE3, which we used to express AimA, does not produce enterobactin. However, in comparison to the wide enterobactin-binding cleft (calyx) of LCN2, the homologous regions of the N- and C-term domains of AimA are much narrower, are occluded by loops or a helix, and lack the enterobactin-interacting residues of LCN (*Figure 3—figure supplement 1A–F*). To test whether AimA binds enterobactin, we grew *E. coli* K12 strain (MG1655) under iron limiting conditions that make it dependent on the siderophore function of its secreted enterobactin, which can be sequestered by exogenously added LCN2 (*Goetz et al., 2002*). As predicted, when we grew *E. coli* K12 in the presence of the iron chelator dipyridyl to limit available iron, we observed a dose dependent inhibition of *E. coli* growth upon addition of purified mouse LCN2 protein (*Figure 3—figure supplement 1G*). In contrast, when we added the same range of concentrations of purified AimA

protein to the *E. coli* cultures, we saw no growth inhibition (*Figure 3—figure supplement 1G*), suggesting that AimA does not bind and sequester enterobactin in a similar manner to LCN2.

## Each individual domain of AimA retains neutrophil regulating capacity

Human LCN2 exists both as a monomer and a homodimer, and while the functional distinction between the two forms is unknown, the homodimer is the major molecular form secreted by neutrophils (*Cai et al., 2010*). Both the C- and N-terminal domains of AimA have structural homology to proteins in the calycin superfamily. Although they share only 17% amino acid identity, a structural overlay between the two domains reveals some structural conservation (*Figure 4A* and *Figure 4—figure supplement 1A*). Because human LCN2 exists as both a monomer and homodimer and the two domains of AimA each have a lipocalin-like fold, we asked if each domain alone was sufficient to

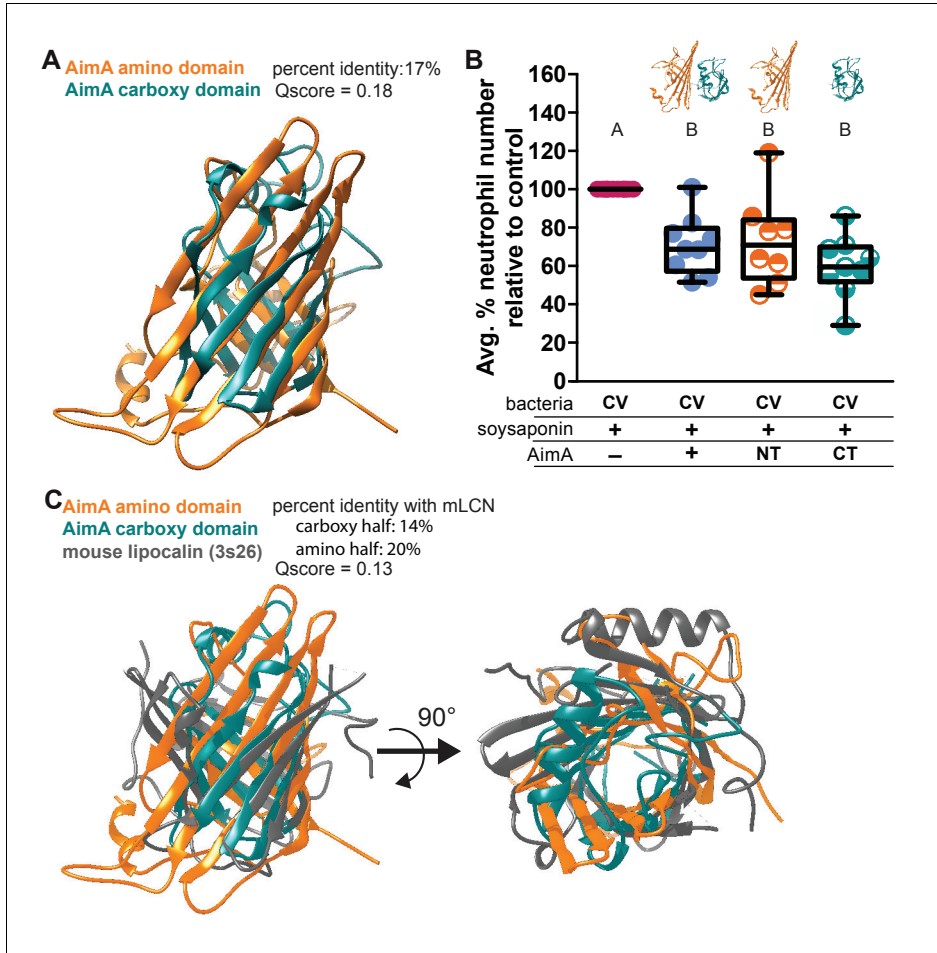

**Figure 4.** Both domains of AimA retain neutrophil modulating function. (**A**) Overlay of C-term and N-term domains of AimA using PDBeFold. (**B**) Conventionally raised (CV) fish fed soysaponin (SS) and treated with either purified full-length AimA or purified N-term (NT) or C-term (CT). Each dot represents the average percent of neutrophil influx in a flask of 15 fish from the average neutrophil influx of a control flask (soysaponin only) of 15 fish. n ≥ 9 flasks from at least three independent experiments. Letters indicate significantly different groups; ANOVA with multiple comparisons. (**C**) Overlay of C- and N-term domains of AimA and mLCN using PDBeFold.
DOI: https://doi.org/10.7554/eLife.37172.016

The following source data and figure supplement are available for figure 4:

**Source data 1.** Both domains of AimA retain neutrophil modulating function.
DOI: https://doi.org/10.7554/eLife.37172.018

**Figure supplement 1.** Structural comparison of N-term and C-term domains of AimA.
DOI: https://doi.org/10.7554/eLife.37172.017

alter the intestinal neutrophil response. The individual domains were not as soluble as full-length AimA, but we were able to crudely purify each domain. Concurrently with soysaponin, we added approximately 100 ng/mL of the domain of interest to zebrafish from 4 to 6 dpf. We found that both the N- and C-term domains alone were sufficient to reduce the neutrophil response in the soysaponin model of intestinal inflammation (*Figure 4B*). An in depth analysis of the structural overlay between the N- and C-term domains revealed that seven out of eight β-strands in the barrels align well; however, only seven total residues are in analogous functional positions across the two domains (*Figure 4—figure supplement 1*). Of these seven residues, two—Val 60/201 and Thr 117/265—stand out as possible candidates to interact with a hydrophobic ligand that could bind inside the barrel cavity. The other five residues may overlap by coincidence or they may be positioned to interact with a promiscuous protein or ligand partner. Furthermore, given the overall structural similarity between the N- and C-term domains, it is possible that critical residues are in a flexible loop region that could become structured upon binding (*Figure 4A* and *Figure 4—figure supplement 1A*). Interestingly, both AimA domains have comparable structural similarity with mLCN2 (*Figure 4C*). While the mLCN2 binding cleft for enterobactin is not conserved in the domains of AimA, the majority of the β-strands in the barrels overlap in all three domains, which suggests these domains could interact with the same binding partner(s) (*Figure 4C*).

## AimA controls host neutrophil response and promotes *Aeromonas* colonization

With a clear understanding that AimA controls the host intestinal neutrophil response, we next asked whether the activity of AimA also increases *Aeromonas* fitness, thus facilitating a mutualistic relationship with the host. We began by asking how prevalent AimA was across bacterial genomes. Knowing that proteins in the calycin superfamily have low sequence conservation, we were not surprised to find AimA homologues by sequence similarity only within the *Aeromonas* genus (*Figure 5—figure supplement 1A*). We found an almost identical homologue (99%) to AimA in a zebrafish intestinal *Aeromonas* isolate, ZOR0001, referred to here as ZF Aer (*Table 1*) (*Stephens et al., 2016*). Some *Aeromonas* species, including *A. veronii* strain Hm21 (*Table 1*), have both AimA and a second copy, which we named AimB. AimB is distantly related to AimA by amino acid sequence conservation (27%), yet a model generated by the Iterative Threading ASSEmbly Refinement (I-TASSER) program of the structure of AimB overlays directly on the structure of AimA (*Figure 5A*) (*Yang et al., 2015*; *Roy et al., 2010*; *Zhang, 2008*). A reexamination of our mass spectrometry analysis of the ΔT2C CFS uncovered peptides corresponding to AimB that were enriched in the sample from ΔT2C compared to the ΔT2 mutant, although the difference between the two samples was less than our original threshold (*Supplemental file 1*).

To determine whether the bioactive protein AimA and its homologue AimB benefit *Aeromonas*, we constructed deletion strains of each gene individually and of both genes in *A. veronii* strain Hm21 (Aer Δ*aimA*, Aer Δ*aimB*, and Aer Δ*A*Δ*B*) and an AimA deletion in the ZF Aer background (ZF Aer Δ*aimA*). All of these mutants displayed normal growth in vitro in rich media (*Figure 5—figure supplement 1B*). Because AimA has structural homology to LCN2, which binds siderophores, and *Aeromonas* species make several siderophores, including enterobactin (*Maltz et al., 2015*), we asked whether AimA functions in iron acquisition for *Aeromonas* by testing the ability of wild type Hm21 and the Aer Δ*A*Δ*B* mutant to grow in iron depleted media containing dipyridyl. We found that the two strains had identical growth curves when grown under iron starvation (*Figure 5—figure supplement 1C*), suggesting that AimA and AimB do not function in iron acquisition for *Aeromonas*.

We next tested whether the strains lacking *aim* genes exhibited growth defects in mono-associations in larval zebrafish. We found that when inoculated into GF larvae at 3 dpf, the ZF Aer Δ*aimA* mutant colonized gnotobiotic zebrafish to lower levels than its wild type counterpart (*Figure 5B*), with the colonization defect becoming apparent at 7 dpf (*Figure 5—figure supplement 2*). We observed a similar colonization defect at 7 dpf with the Aer Δ*A*Δ*B* mutant relative to the wild-type strain (*Figure 5C*). However, the single gene deletion strains colonized to wild-type levels, suggesting that AimA and AimB provide redundant functions for *Aeromonas* growth within the larval intestine. We note that Aer ΔT2 did not exhibit a colonization defect (*Figure 1—figure supplement 1*) similar to the Aer Δ*A*Δ*B* mutant, which could be due to other phenotypic consequences of its altered secretion profile.

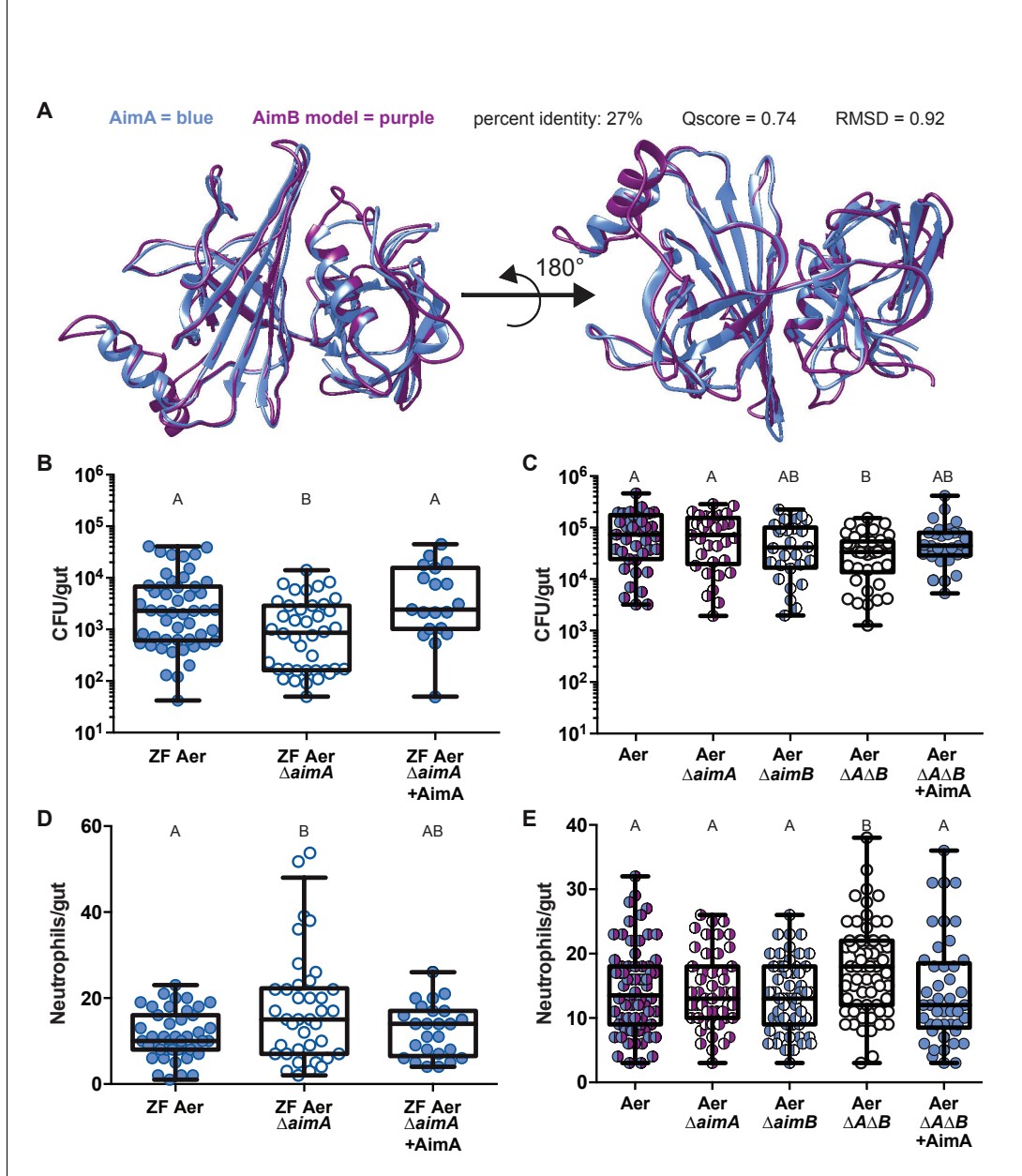

**Figure 5.** AimA reduces neutrophil influx and promotes colonization of *Aeromonas*. (A) Structural overlay (PDBeFold) of AimA (blue) and the model of AimB (purple) generated by I-TASSER using AimA as a threading structure. (B) Colonization level of wild-type ZF *Aeromonas* and ZF Aer ΔAimA. The colonization defect of ZF Aer ΔAimA is rescued by treatment with 100 ng/mL purified AimA. (C) Colonization level of wild-type *Aeromonas*, ΔAimA, ΔAimB, and ΔAΔB. Each of the single mutants colonizes as well as wild type, while the double mutant has a significantly reduced colonization level. This phenotype is rescued by treatment with 100 ng/mL purified AimA. (D) Intestinal neutrophil response to ZF Aer and the colonization defect of ZF Aer ΔAimA. (E) Intestinal neutrophil response to wild-type *Aeromonas*, ΔAimA, ΔAimB, and ΔAΔB. Each of the single mutants induces a similar neutrophil response to wild type, while the double mutant induces significantly greater response. This phenotype is rescued by treatment with 100 ng/mL purified AimA for both *Aeromonas* isolates. For all graphs, each dot represents one fish; n ≥ 23 from at least three independent experiments. Letters indicate significantly different groups, ANOVA with multiple comparisons.
DOI: https://doi.org/10.7554/eLife.37172.019

The following source data and figure supplements are available for figure 5:

**Source data 1.** AimA reduces neutrophil influx and promotes colonization of *Aeromonas*.
DOI: https://doi.org/10.7554/eLife.37172.022

**Figure supplement 1.** Homologues to AimA are found across the *Aeromonas* genus.
DOI: https://doi.org/10.7554/eLife.37172.020

*Figure 5 continued on next page*

*Figure 5 continued*

**Figure supplement 1—source data 1.** Homologues to AimA are found across the *Aeromonas* genus.

DOI: https://doi.org/10.7554/eLife.37172.023

**Figure supplement 2.** ZF Aer Δ*AimA* suffers the greatest colonization defect seven dpf.

DOI: https://doi.org/10.7554/eLife.37172.021

**Figure supplement 2—source data 1.** ZF Aer D*AimA* suffers the greatest colonization defect 7 dpf.

DOI: https://doi.org/10.7554/eLife.37172.024

We next tested whether colonization defects of the *aim* deficient strains could be rescued by supplementation of the flask media with purified AimA protein. Remarkably, addition of purified AimA protein at a concentration of 100 ng/mL to the flask water from 4 to 7 dpf completely restored the colonization efficiency of the *aim* deficient strains (*Figure 5B,C*).

Because purified AimA reduces the intestinal neutrophil response (*Figure 3A,C*), we hypothesized that the *aim* deletion strains would induce more inflammation, and thus upon mono-association could create a less hospitable intestinal environment for *Aeromonas* colonization. To test our hypothesis, we quantified intestinal neutrophil numbers in response to Δ*aim* mono-associations at 7 dpf, corresponding to the time point when the bacterial colonization deficit for ZF Aer Δ*aimA* was apparent (*Figure 5—figure supplement 2*). We found that zebrafish mono-associated with ZF Aer Δ*aimA* exhibited significantly higher intestinal neutrophil counts as compared with those colonized with wild-type ZF Aer (*Figure 5D*), despite carrying a lower bacterial load (*Figure 5B*). Similarly, zebrafish colonized with the double mutant, Aer Δ*A*Δ*B*, had significantly higher neutrophil counts than fish colonized with wild-type Aer (*Figure 5E*), despite being colonized at a lower level (*Figure 5C*). Notably, the Aer Δ*aimA* and Aer Δ*aimB* single mutants, which did not exhibit decreased colonization in mono-associations, also did not induce higher intestinal neutrophil numbers (*Figure 5E*). For both *Aeromonas* lineages, when exogenous AimA was added to the flasks of fish mono-associated with inflammation-inducing Δ*aim* strains, the protein reduced neutrophil numbers back to normal levels (*Figure 5D,E*) and at the same time restored colonization levels (*Figure 5B,C*).

## AimA promotes host survival of inflammatory challenge

We previously showed that resident zebrafish bacterial species can vary dramatically in their per capita impact on host response to bacterial colonization (*Rolig et al., 2015*). By extrapolating the number of neutrophils elicited per $10^4$ colonizing bacteria, we calculated that Aer Δ*A*Δ*B* is much more immuno-stimulatory that the wild-type strain, recruiting nearly twice the number of neutrophils, on average, for the same number of bacteria (*Figure 6A*). This suggests that the Aim proteins function to reduce *Aeromonas*' immune stimulation, allowing it to reach high colonization density without eliciting a strong inflammatory response.

Knowing that loss of the Aim proteins resulted in both a significantly increased per capita effect and intestinal neutrophil response, we explored whether the increased neutrophil response led to other downstream consequences for the host by monitoring the survival rate of mono-associated fish. By 72 hr post-inoculation, the survival rate of zebrafish colonized with wild-type Aer was 92% (n = 163). We observed a significant decline in the survival of fish mono-associated with *Aeromonas* lacking Aim genes [Aer Δ*aimA* (n = 139), Aer Δ*aimB* (n = 174), and Aer Δ*A*Δ*B* (n = 148)], with survival rates of 64%, 73%, and 55%, respectively (*Figure 6B*). Remarkably, this decreased survival rate was rescued back to 90% by the presence of purified AimA protein (n = 60; *Figure 6B*). These data demonstrate that the Aim proteins act to promote both bacterial colonization and host survival, identifying AimA as a key mediator of host-bacterial mutualism.

To test whether AimA benefits both the bacteria and the host by controlling the neutrophil response to bacterial colonization, we evaluated the importance of Aim proteins in a colonization model lacking intestinal neutrophils. We reasoned that in the absence of intestinal inflammation, bacteria possessing or lacking Aim proteins would have a similar capacity to survive in the intestine. For these experiments, we used *myd88*$^{-/-}$ mutant fish lacking the adaptor protein for inducing many toll-like receptor and IL-1 dependent pro-inflammatory pathways (*Larsson et al., 2012*); these fish have a severely attenuated intestinal neutrophil response (*Figure 6C*) that is indistinguishable from GF wild-type zebrafish (*Figure 1A*), as we have reported previously (*Bates et al., 2007*; *Burns et al.,*

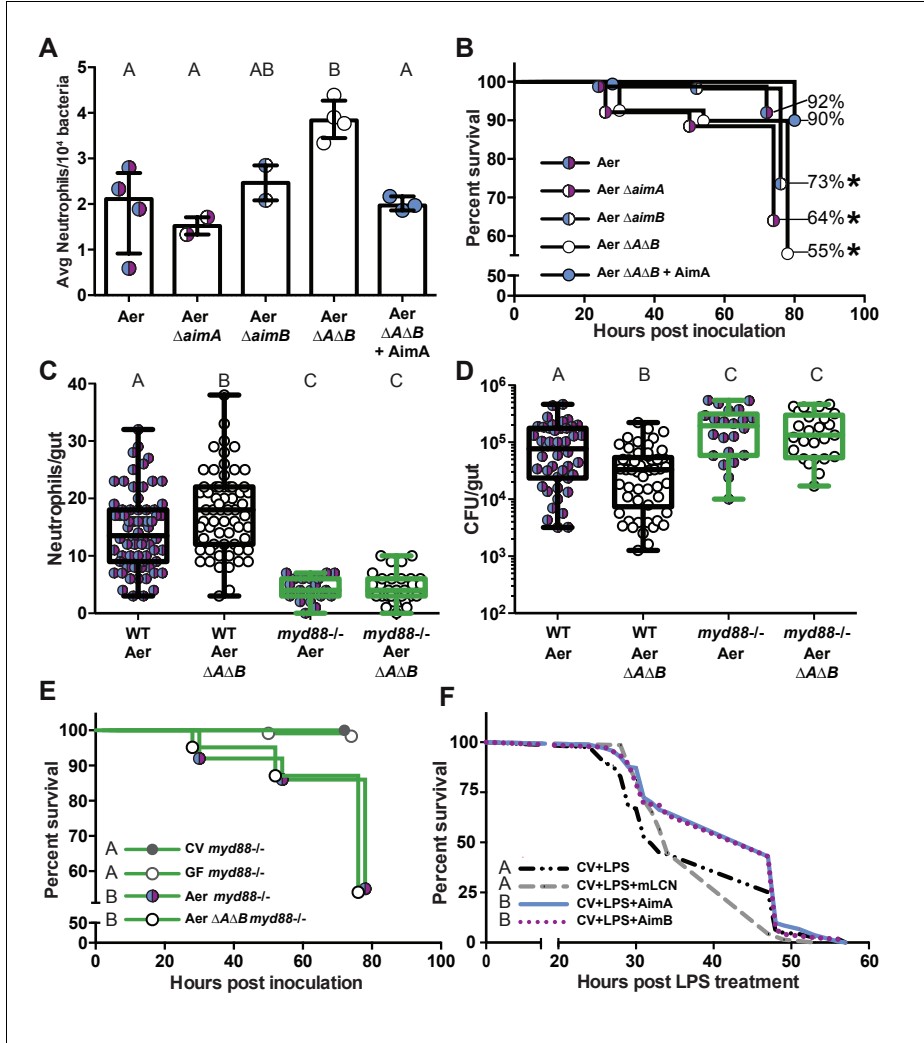

**Figure 6.** The increased neutrophil response to *Aeromonas* ΔAΔB causes decreased survival rate. (**A**) The per capita effect of wild-type *Aeromonas*, ΔAimA, ΔAimB, and ΔAΔB. Each dot represents the average neutrophil response from a flask of 15 fish divided by the average colonization level from a flask of 15 fish, normalized to $10^4$. (**B**) Survival curve of zebrafish mono-associated with wild-type *A. veronii* (N = 163), ΔAimA (N = 139), ΔAimB (N = 174), ΔAΔB (N = 148), and ΔAΔB + AimA (N = 60). *indicates significant difference from the survival curve with wild-type *Aeromonas*, Mantel-Cox test. (**C**) *myd88*[-/-] transgenic fish lack a neutrophil response to bacteria. Each dot represents one fish. (**D**) ΔAΔB colonization is rescued to wild-type colonization levels in *myd88*[-/-] transgenic fish. Further, both wild-type *Aeromonas* and ΔAΔB reach significantly higher colonization levels in the *myd88*[-/-] transgenic fish compared to wild-type fish, indicating that the innate immune response limits commensal bacterial growth. (**E**) Survival curves of *myd88*[-/-] zebrafish over a 3 day infection. GF *myd88*[-/-] (N = 45), CV *myd88*[-/-] (N = 120), Aer *myd88*[-/-] (N = 100), ΔAΔB *myd88*[-/-] (N = 124). (**F**) Survival curve of conventionally raised (CV) fish treated with LPS (N = 163), with LPS and AimA (N = 163), with LPS and AimB (N = 121), or with LPS and mLCN (N = 69). Letters next to key indicate significant difference between the survival curves, Mantel-Cox test. For all graphs, letters indicate significance by ANOVA with multiple comparisons.

DOI: https://doi.org/10.7554/eLife.37172.025

The following source data is available for figure 6:

**Source data 1.** The increased neutrophil response to *Aeromonas* ΔAΔB causes decreased survival rate.
DOI: https://doi.org/10.7554/eLife.37172.026

*2017*). We found that both Aer ΔAΔB and wild-type Aer colonized *myd88*^-/- fish to equivalent levels, which were significantly higher than in wild-type fish (*Figure 6D*). These results suggest that the decreased colonization of Aer ΔAΔB compared to wild-type *Aeromonas* in wild-type fish is due to the increased inflammatory response to the mutant bacteria, and moreover suggest that the hosts' innate immune response contributes to limiting bacterial colonization. We further found that *myd88*^-/- fish inoculated with wild-type *Aeromonas* or Aer ΔAΔB died at the same rate (*Figure 6E*), suggesting that the increased inflammatory response of wild-type fish colonized with Aer ΔAΔB compared to wild-type *Aeromonas* accounts for the decreased fish survival rate with the mutant bacteria. Notably, the *myd88*^-/- fish mono-associated with either wild-type or Aer ΔAΔB perished at higher rates than wild type fish, indicating that there is a host fitness cost to failing to control *Aeromonas* levels.

Given the increase in bacterial colonization and fish death in the *myd88*^-/- fish, regardless of the colonizing strain, we hypothesized that these fish may be dying of septic shock as a result of bacterial overgrowth. Thus we asked whether AimA could delay death as a result of another zebrafish model of bacterial septic shock, which is induced by LPS intoxication (*Bates et al., 2007*; *Philip and Wang, 2017*). To test this, we treated fish with 100 ng/mL AimA on 4 dpf and then challenged them with 600 μg/mL LPS on 5 dpf and tracked their survival over the subsequent 60 hr. The median survival time post LPS treatment was 32 hr, which we found was extended to 47 hr in the presence of AimA (*Figure 6F*). Using the same treatment paradigm, we found that purified AimB also extended the median survival of LPS-exposed zebrafish to 47 hr (*Figure 6F*). In contrast, treatment with mLCN2 had no effect on zebrafish survival (median survival 34 hr, *Figure 6F*). Thus, AimA and AimB, but not mLCN2, significantly extend the survival of zebrafish in an LPS intoxication model.

# Discussion

## A universal feature of animals is their co-existence with resident microbial communities.

Appropriate modulation of host immune responses to resident microbes is critical for maintaining health, with too muted a response leaving hosts susceptible to microbial overgrowth (*Dukowicz et al., 2007*; *Rolig et al., 2017*) and too aggressive a response wreaking havoc on both host tissue and resident microbiota (*Huttenhower et al., 2014*). Fitness effects of host-bacterial interactions are often considered from the host perspective, yet it is essential to examine both sides of these partnerships to understand how these interactions evolve and persist. Here we describe the discovery of AimA, a bacterial immunomodulatory protein secreted by *Aeromonas* that is required during bacterial colonization of larval zebrafish to prevent an intestinal inflammatory response that is detrimental to both partners. We determined the molecular structure of AimA and show that it consists of two structurally similar lipocalin-like domains, each of which acts to suppress intestinal inflammation. We demonstrate that purified AimA protein is protective against host intestinal inflammation and LPS intoxication and simultaneously rescues *Aeromonas* growth in the context of inflammation. When the inflammatory response is attenuated in *myd88* deficient hosts, then the bacterial colonization advantage associated with AimA production is eliminated, indicating that it is AimA's anti-inflammatory activity that benefits *Aeromonas*.

All resident bacteria produce microbial associated molecular patterns (MAMPs) that are inherently pro-inflammatory, as we have shown in zebrafish larvae for LPS, which elicits intestinal neutrophil influx, proinflammatory cytokine gene expression, and endotoxemia in a dose-dependent manner (*Bates et al., 2007*). However, our analysis of dose response curves to resident bacteria revealed that few zebrafish microbiota members elicit dose-dependent inflammation (*Rolig et al., 2015*; *Rolig et al., 2017*). We did identify a few examples of bacteria with potent and dose-dependent inhibitory effects on intestinal neutrophil influx. These dose-dependent inhibitory effects were exploited to identify an immunoregulatory *Shewanella* strain that produces a secreted, heat-labile, anti-inflammatory factor (*Rolig et al., 2015*) and an *Escherichia* strain that reduces intestinal inflammation through a secretion-independent mechanism (*Rolig et al., 2017*). However, the majority of zebrafish gut bacteria we surveyed, including *Aeromonas* species, showed no correlation between their bacterial load and the number of neutrophils they elicited (*Rolig et al., 2015*; *Rolig et al., 2017*), suggesting that most gut bacteria posses both pro-inflammatory products, like LPS, along

with neutralizing mechanisms to counteract the effects on their hosts. AimA, which decreases intestinal neutrophil influx and septic shock elicited by both *Aeromonas* and LPS, represents such a neutralizing activity. Future studies will determine whether AimA acts primarily by reducing the production of pro-inflammatory cytokines that recruit neutrophils to the intestine or by inhibiting the responsiveness of neutrophils to such cues.

Sequence-based searches for AimA-related proteins recovered only other *Aeromonas* homologues, including the highly divergent but functionally redundant AimB. Structural determination of AimA, however, revealed two linked domains of the lipocalin superfamily. Lipocalin domains are found across the tree of life, with bacterial lipocalin fold-containing proteins being only more recently identified (*Bishop, 2000*). In eukaryotes, proteins with lipocalin folds commonly bind small hydrophobic molecules such as odorant molecules, retinol, and fatty acids (*Flower, 1996*), but proteins with a lipocalin fold, especially those found in prokaryotes, can also take on a variety of other functions, such as inducing apoptosis (*Tavares and Pathak, 2017*), binding extracellular matrix (*Parker et al., 2016*), and binding hydrophobic antibiotics and antimicrobials (*El-Halfawy et al., 2017*; *Sisinni et al., 2010*).

We were particularly intrigued by AimA's structural similarity to mammalian LCN2, a known regulator of intestinal inflammation (*Moschen et al., 2017*; *Xiao et al., 2017*). LCN2 was originally identified and purified from neutrophil granules and shown to play a critical role in neutrophil extravasation and migration—processes that AimA may also modulate. LCN2 was subsequently shown to be a sensitive fecal biomarker of intestinal inflammation in humans and mice (*Chassaing et al., 2012*). Inflammation-associated induction of LCN2 appears to be protective against pathology because *Lcn2* deficient mice are sensitized to developing colitis (*Toyonaga et al., 2016*; *Moschen et al., 2016*; *Singh et al., 2016*). Further, *Lactococcus lactis* engineered to overexpress mLCN2 confers greater protection against colitis than a control strain of *L. lactis* in a mouse model of acute colitis (*Saha et al., 2017*). While it is clear that LCN2 protects against inflammation, the mechanisms by which it does so are not well understood. One possibility is that mLCN2's capacity to sequester bacterial enterobactin-like siderophores may cause beneficial shifts in intestinal microbiota composition and reduce the toxic effects of iron in the inflamed environment (*Moschen et al., 2016*). Our data, however, suggest that AimA confers protection against inflammation by a distinct mechanism, because we found no evidence that AimA binds enterobactin or contributes to *Aeromonas* survival in iron limiting conditions. To test for a possible overlap between LCN2 and AimA function, we asked whether addition of mLCN2 to a model of soysaponin-induced intestinal inflammation would alter AimA's efficacy. Indeed, the presence of mLCN2 inhibited the ability of AimA to reduce the neutrophil response. This result may indicate that the two proteins compete by acting on the same pathway governing neutrophil behavior. For example, AimA and mLCN2 may compete directly for binding to the same host receptor. LCN2 binds at least two known mammalian receptors, megalin and LCN2R, which have homologues in zebrafish (*Kim et al., 2011*), although the binding to these receptors has not been directly connected to the influence of LCN2 on neutrophil behavior (*Moschen et al., 2017*; *Cabedo Martinez et al., 2016*). Another possibility is that mLCN2 interferes with AimA's activity by binding directly to AimA. mLCN2 is known to exist both as monomers and as dimers (*Axelsson et al., 1995*; *Perduca et al., 2001*; *Niemi et al., 2015*), and while little is known about the conditions under which each form is active, the dimeric form is associated with neutrophils (*Cai et al., 2010*). In AimA, the two lipocalin-like domains are tethered, but we found that each structural half can function independently to inhibit inflammation. Future biochemical studies will determine whether either of AimA's lipocalin domains can homodimerize or bind to other lipocalin proteins and which oligomeric states of AimA confer its immune modulatory function.

Our analysis of *aimA* and *aimB* deficient *Aeromonas* revealed that the proteins' anti-inflammatory activity on the host is inextricable from the fitness advantage it confers to the bacteria. In this regard, AimA represents a new class of bacterial effector proteins, which we refer to as mutualism factors. Whereas virulence factors promote the fitness of pathogens to the detriment of their hosts and are revealed by the decreased fitness of the pathogen mutants coupled to the increased fitness of the host, a mutualism factor like AimA confers mutual benefits, and both partners suffer when it is removed from the host-microbe symbiosis. Another example of a mutualism factor is the zwitterionic polysaccharide PSA produced by *Bacteroidetes fragilis*, which colonizes the murine colon. PSA induces T regulatory cells (T regs), which benefit its murine host by preventing excessive inflammation.

The T regs in turn contribute to *B. fragilis* colonization, as demonstrated by the fact that in the absence of T regs, due either to genetic ablation or absence of PSA-induction, *B. fragilis* suffers a colonization defect (*Round et al., 2011*).

Given the constraints of bacterial-host coexistence in the vertebrate intestine, we speculate that intestinal microbiota are rich sources of novel anti-inflammatory factors that enable resident microbes to colonize this tissue at high density without eliciting excessive inflammation. Such anti-inflammatory factors offer promise for useful therapeutic applications, and the high throughput gnotobiotic larval zebrafish system provides an experimentally tractable platform for their discovery. These factors, however, may represent novel biological molecules that defy easy functional classification based purely on their gene sequences or the gene sequences of their biosynthetic machinery, thus additional approaches, such as protein structure determination, will be needed to further advance our understanding of their mechanisms of action. In the case of *Aeromonas* AimA, the crystal structure revealed it to belong to a large family of lipocalin proteins, which includes the mammalian immunomodulatory protein LCN2. Our discovery of AimA raises the possibility that AimA and LCN2 have overlapping functions as mutualism factors in the vertebrate intestine even though they are provisioned by different members of the partnership. Given the diversity of the microbial communities that are part of the complex host-microbe ecosystem, AimA is likely only the first of many unique bacterial proteins that facilitate mutualistic interactions critical for both host and resident microbes to thrive.

# Materials and methods

**Key resources table**

| Reagent type (species) or resource | Designation | Source or reference | Identifiers | Additional information |
|---|---|---|---|---|
| Strain, strain background (*Escherichia coli*) | BL21 DE3 | New England Biolabs | C2527 | |
| Strain, strain background (*E. coli*) | K12 strain (MG1655) | doi: 10.1128/JB.188.3.928–933.2006 | | Dr. Matthew Mulvey, Univ. of Utah |
| Strain, strain background (*Aeromonas veronii*) | HM21S; Aer; Aeromonas Hm21; Hm21 | doi:10.1128/AEM.01621–10 | | Parent strain, SmR |
| Strain, strain background (*A. veronii*) | HE-1095; Aer DT2 | doi:10.1128/AEM.01621–10 | | Hm21S::interrupted exeM mTn5 KmR SmR |
| Strain, strain background (*A. veronii*) | HEC-1344; Aer DT2C | doi:10.1128/AEM.01621–10 | | HE-1095::Tn7 containing TpR exeMN + promoter region |
| Strain, strain background (*A. veronii*) | ASRC7; Aer D*aimA* | this study | | Hm21S aimA::cmR |
| Strain, strain background (*A. veronii*) | ASRD5; Aer D*aimB* | this study | | Hm21S D*aimB* |
| Strain, strain background (*A. veronii*) | ASRD4; Aer D*ADB* | this study | | Hm21S *aimA::cm*$^R$; D*aimB* |
| Strain, strain background (*A. veronii*) | ZOR0001; ZF Aer | doi:10.1038/ismej.2015.140 | | Zebrafish *Aeromonas* isolate |
| Strain, strain background (*A. veronii*) | ZOR0001; ZF Aer D*aimA* | this study | | ZOR0001 *aimA::cmR* |
| Genetic reagent (*Danio rerio*) | AB x Tu strain; wild type zebrafish | UO Zebrafish facility | | |
| Genetic reagent (*Danio rerio*) | *myd88*-/- | PMID: 28973938 | | |
| Genetic reagent (Danio rerio) | Tg(BACmpx:GFP) i114; mpx:GFP | doi:10.1182/blood -2006-05-024075 | | |
| Recombinant DNA reagent | pET21B | Genscript | | |

*Continued on next page*

*Continued*

| Reagent type (species) or resource | Designation | Source or reference | Identifiers | Additional information |
|---|---|---|---|---|
| Chemical compound, drug | LPS, *E. Coli* O111:B4 | Sigma | L2630 | |
| Peptide, recombinant protein | Lipocalin 2, LCN2, Siderocalin | Biolegend | 588002 | |
| Peptide, recombinant protein | AimA | this study | NCBI hypothetical protein WP_021230730.1 | see Materials and Methods |
| Peptide, recombinant protein | AimB | this study | NCBI hypothetical protein WP_021230165.1 | see Materials and Methods |
| Peptide, recombinant protein | mLCN2 | this study | NCBI gene NM_008491.1 | see Materials and Methods |

## Gnotobiotic zebrafish husbandry

All zebrafish experiments were performed following protocols approved by the University of Oregon Institutional Animal Care and Use Committee and followed standard zebrafish protocols (*Westerfield, 2000*). Conventionally-raised (CV) wild-type (AB x Tu strain), Tg(BACmpx:GFP)i114 (referred to as mpx:GFP) (*Renshaw et al., 2006*) and myd88$^{-/-}$ (*Burns et al., 2017*) were maintained as described (*Westerfield, 2000*). Zebrafish embryos were derived germ free (GF) as previously described (*Melancon et al., 2017*). Subsequently, 15 GF embryos were transferred to sterile tissue culture flasks (25 cm$^2$, Techno Plastic Products, Trasadingen, Switzerland) with 15 mL embryo medium (EM) (*Melancon et al., 2017*). Monoassociated zebrafish were generated by inoculating flasks with 4 dpf GF zebrafish with 10$^6$ colony forming units (CFU)/mL of each bacterial strain. GF flasks were chosen at random for their respective treatment, and the researcher was blinded to the treatment group until after data collection. Bacteria used for monoassociations were zebrafish isolate *Aeromonas* ZOR0001 (*Stephens et al., 2016*), *Aeromonas* ZOR0001ΔaimA, *Aeromonas* ZOR0001 ΔaimA complement, *Aeromonas* HE-1095 (ΔT2) (*Maltz and Graf, 2011*), *Aeromonas* HEC-1344 (ΔT2C) (*Maltz and Graf, 2011*), Aeromonas Hm21S (*Graf, 1999*), Hm21S ΔaimA, Hm21S ΔaimB, and HM21S ΔaimAΔaimB. Control CV fish were prepared GF, as above, and inoculated with 1 mL of fish facility water on 4 dpf. All manipulations to the GF flasks were performed under a class II A/B3 biological safety cabinet. The flasks were kept at 28° C until analysis of fluorescent myeloperoxidase positive (MPX+) cells on 7 dpf.

*Aeromonas* mutants were created using plasmids and protocols as described in *Wiles et al. (2018)*. Briefly, *Aeromonas* mutants were constructed using homologous recombination of 1 kb regions upstream and downstream of the *aimA* gene to replace *aimA* with a chloramphenicol resistance gene. The *aimB* mutant was constructed in a similar manner, except as a markerless deletion by first generating a merodiploid strain.

## Histology and quantification of neutrophils

Zebrafish larvae were fixed in 4% paraformaldehyde (PFA) overnight. Whole larvae were stained with Myeloperoxidase kit (Sigma) following the manufacturer's protocol and processed and analyzed as previously described (*Bates et al., 2007*), except that MPO +cells were quantified in whole dissected intestines rather than tissue sections. For analysis of neutrophils in *mpx:GFP* fish, GFP +cells in the intestine were quantified as previously described (*Rolig et al., 2015*). Briefly, the mpx:GFP zebrafish were anesthetized in Tricaine (Western Chemical, Inc., Ferndale, WA) and mounted in 4% methylcellulose (Fisher, Fair Lawn, NJ), and their intestines were sterilely dissected. The number of GFP-positive cells was quantified visually for each fish using a fluorescent microscope (SteREO Discovery.V8, Zeiss).

## Microbiota quantification

To determine the CFU/intestine, dissected zebrafish intestines were placed in 100 μL sterile EM, homogenized in a bullet blender (Next Advance), diluted, and cultured on tryptic soy agar (TSA, BD, Sparks MD). The TSA plates were incubated at 30° C overnight and then colonies were counted.

## Concentration and fractionation of cell-free supernatant

*Aeromonas* HEC-1344 or *E. coli* BL21 were grown overnight to stationary phase. Then 500 µL of the overnight culture was used to inoculate a 50 mL culture, which was kept shaking at 30° C for 2 hr, until the bacteria had grown through exponential and into stationary phase (OD of approximate 0.6). To overexpress proteins of interest, 1 mM of IPTG was added to the cultures during exponential growth phase and allowed to grow for an additional 2 hr at 30° C. To prepare the CFS, the 50 mL cultures were centrifuged at 7000 x g for 10 min at 4° C. Subsequently, the supernatant was filtered through a 0.22 µm sterile tube top filter (Corning Inc., Corning, NY). The sterile supernatant was concentrated at 4° C for 1 hr at 3000 x g with a centrifugal device that has a 10 kDa weight cut off (Pall Life Sciences, Ann Arbor, MI). The concentration of the supernatant was determined with a Nanodrop and inoculated into the flasks at a final concentration of 500 ng/mL.

Ammonium sulfate fractionation experiments were done as previously described (*Hill et al., 2016*). Briefly, unconcentrated CFS from a 50 mL overnight culture of *Aeromonas* HEC-1344 was fractionated by slowly adding cold 100% ammonium sulfate until solutions reached 20%, 40%, and 60% ammonium sulfate. Precipitated proteins were collected by centrifugation at 4° C, 15000 g for 15 min. Recovered proteins were resuspended in cold EM, dialyzed for 2–3 hr at 4° C, then added to mono-associated fish flasks at a concentration of 500 ng/mL.

## Mass spectrophotometry

The protein constituents of concentrated cell free supernatants from *Aeromonas* HE-1095 and *Aeromonas* HEC-1344 (*Maltz and Graf, 2011*) were determined by analysis of peptide MS/MS spectra at the Proteomics Shared Resource Facility at Oregon Health and Sciences University in Portland, Oregon.

## Inflammatory assays

Soysaponin (Sigma) was mixed with Zieglers fish food at a concentration of 0.3%. Ten CV zebrafish were maintained 10 mL EM in 60 × 15 mm petri dishes. Larval fish were fed once daily from 4 dpf to 6 dpf. During each feeding, the larvae had access to the food for 3–4 hr before being washed into fresh EM. For experiments with mLCN2, recombinant mouse LCN2 (Biolegend) was added to the fish EM at a concentration of 100 ng/mL after the soysaponin feeding on 4 dpf and 5 dpf after the fish were moved into fresh EM.

To induce LPS intoxication, CV zebrafish were treated with 600 µg/mL LPS (Sigma) on 5 dpf and monitored for survival for the following 2 days. In additional treatment groups, 100 ng/mL AimA, AimB, or mLCN (purified as described below) were added on 4 dpf.

## AimA, AimB, and mLCN2 protein purification

The *aimA* gene was PCR amplified from gDNA excluding the 5' 66 nucleotide (22 amino acid) secretion signal and cloned into pET21b using NdeI and XhoI restriction sites. The resultant gene product (NCBI hypothetical protein WP_021230730.1) expressed well in *E. coli* BL21 DE3 as a C-terminal 6X His tagged protein of 300 amino acids long (including His tag and linker).

The *E. coli* culture was grown at 37° C until OD600 0.4–0.6, then moved to 30° C and induced with 1 mM IPTG for 3–4 hr. All subsequent steps were performed at 4° C. 1–2 L of pelleted *E. coli* cells were lysed in lysis buffer (50 mM HEPES pH 7.9, 300 mM NaCl, 10 mM imidazole and 5 mM BME), sonicated, and debris pelleted. The supernatant was washed over 5 mL Ni-NTA resin in a gravity column that was pre-washed with lysis buffer. The resin was washed with 15X bed volume of lysis buffer, then 10X bed volume of lysis buffer with 30 mM imidazole, 10X bed volume of lysis buffer with 50 mM imidazole and finally eluted with 3x - 5x bed volume of lysis buffer plus 100–300 mM imidazole. The high absorbance (280 nm wavelength) fractions were pooled and dialyzed overnight into 150 mM NaCl, 50 mM HEPES pH 7.9 and 5 mM BME, concentrated, flash frozen in liquid nitrogen, and stored at −80° C.

The *aimB* gene, excluding the 5' 57 nucleotide (19 amino acid) secretion signal, was cloned into pET21b using NdeI and XhoI restriction sites (GenScript). The resulting gene product (NCBI hypothetical protein WP_021230165.1) expressed well in *E. coli* BL21 DE3 as a C-terminal 6X His tagged protein of 321 amino acids long (including His tag and linker). AimB protein was expressed and purified similarly to AimA, above, with some changes. The lysate was run over HisTrap HP 5 mL

column (GE healthcare) using an AKTA FPLC (GE healthcare) and eluted with a gradient from 10 to 500 mM imidazole. Fractions containing the purest AimB were pooled, dialyzed (150 mM NaCl, 50 mM HEPES pH 7.9 and 5 mM BME) and concentrated. Due to the difficulty of separating away contaminating *E. coli* proteins, AimB was only crudely purified to >50% pure by SDS PAGE.

The mouse *lipocalin2* gene (NCBI gene NM_008491.1), excluding the 5' 63 nucleotide (21 amino acids) secretion signal, was cloned into pET21b (GenScript), using Nde1 and Xho1 restriction sites. Mouse LCN2 with a C-term 6X His tag expressed well in *E. coli* BL21 DE3 using 0.25 mM IPTG at 18° C overnight. mLCN2 was purified using Ni-NTA agarose resin (Qiagen) in a gravity column with increasing concentrations of imidazole from 10 to 30 mM, and eluted with 50 mM imidazole. The resultant 22 kDa protein (189 amino acids including His tag and linker) was dialyzed into 300 mM NaCl, 50 mM potassium phosphate pH 7 and 5 mM BME. The dialyzed protein was concentrated and confirmed to be >95% pure by SDS PAGE.

## AimA protein crystallization

Purified AimA with C-terminal 6x His tag was concentrated to 10.9 mg/mL and set up in hanging drops as 1 μL protein: 1 uL well solution at room temperature. AimA crystallized in thick hexagons in 3.5 M sodium formate, 75 mM NaCl, 25 mM HEPES pH 7.9 and 2.5 mM BME. One to two weeks after the crystals grew, they were transferred to wells containing 3.8 M sodium formate to toughen them up for approximately one week. The heavy atom derivative crystals were then transferred to drops with 3.8 M sodium formate, 0.5 M NaI (for iodide data set) and 15% glycerol (as a cryoprotectant) for several hours before being scooped and flash frozen in liquid nitrogen for data collection at the Advanced Light Source in Berkeley, CA, beamline 5.0.2 using the Pilatus detector at a wavelength of 1.0 Å. The native crystals were transferred to 3.8 M sodium formate and 15% PEG 200 briefly, then flash frozen in liquid nitrogen for data collection as described for the iodide soaked crystals.

## Structure determination of AimA

### Data processing

The heavy atom derivative was solved by the single-wavelength anomalous diffraction (SAD) method from a single crystal derivatized with I⁻, with data collected at wavelength $\lambda = 1$ Å. The data set was integrated and scaled to resolution 2.7 Å using HKL3000 (*Minor et al., 2006*) with the merging analysis indicating the P622 space group. Although data were isotropic, diffraction spots were smeared in a manner indicating the presence of order-disorder. In addition, the scaling B-factor increase of ~40 Å$^2$, which is equivalent to a dose of ~40 MGy, indicated severe radiation damage. Therefore, it was necessary to apply the 'automatic corrections' computational procedure to optimize the error model (*Borek et al., 2010*; *Borek et al., 2013*; *Borek et al., 2007*), and this was essential for the success of the experimental phasing described below.

The estimated level of anomalous signal was ~3.6% of the native intensity. The search for heavy atom positions was performed to a resolution of 3.7 Å. The 30 positions of I⁻ were identified using SHELXC/D (*Sheldrick, 2008*), run within HKL3000, with correlation coefficients: $CC_{All} = 41.4\%$, $CC_{Weak} = 16.1\%$. The handedness of the best solution was determined with SHELXE. The heavy atom positions were refined to 2.7 Å with MLPHARE (*Otwinowski, 1991*), with the final Figure of Merit (FOM) reaching 0.14 for all observations. Solvent flattening was performed by DM (*Cowtan and Main, 1998*). The procedure produced an interpretable electron density map that was used for iterative automatic model building with BUCCANEER, Coot and REFMAC (*Cowtan, 2006*; *Emsley and Cowtan, 2004*; *Murshudov et al., 1997*; *Murshudov et al., 1999*) – all run within HKL3000 with the 'HKL Builder' option. That procedure resulted in 100% of the model being built with 90% of the side chains docked. At this point, the R and R-free factors were ~27% and~35%, respectively.

Two native AimA datasets from separate crystals were indexed and integrated with iMosflm 7.2.1 (*Battye et al., 2011*) and scaled using SCALA (*Collaborative Computational Project, Number 4, 1994*). The two datasets were found to be isomorphous, and were combined using POINTLESS (*Collaborative Computational Project, Number 4, 1994*). The high-resolution cutoff was determined by the method of Karplus and Diederichs (*Karplus and Diederichs, 2012*) using a $CC_{1/2}$ of >0.3 and completeness of >50% in the highest resolution shell. This method has been utilized in

numerous other studies (*Perkins et al., 2016*; *Evans and Murshudov, 2013*; *Kern et al., 2013*) and has been cited over 900 times since its publication in *Science* in 2012. Using this criteria the correlation between two halves of the data are used to determine the point at which signal falls away into noise (*Evans and Murshudov, 2013*), and $R_{merg}$ values can rise to values much higher than what has traditionally been thought of as allowable. Using this strategy we were able to extend the resolution from 2.9 Å (where the data would have been cut based on $R_{merg}$ ~0.6) to 2.3 Å. To further test the validity of using this noisy high-resolution data to refine the model, a series of paired refinements were conducted (*Karplus and Diederichs, 2012*). The model was first refined using data out to 2.9, 2.7, 2.5, or 2.3 Å and then, since R values are only comparable when calculated at the same resolution (*Karplus and Diederichs, 2012*), R and $R_{free}$ were calculated for each refined model at 2.9 Å (*Figure 2—figure supplement 2*). The extra resolution improves both R (higher value) and $R_{free}$ (lower value), showing that the model is improved in predictive quality and is less overfit using the extended resolution cutoff. In this case $CC_{1/2}$ remains quite high at 0.9 in the high-resolution shell, and the $<I/\sigma>$ at 1.4 is not far below a traditional cutoff of 2.0. Data statistics are summarized in *Table 2*.

## Structure refinement

Manual model building was performed using Coot 0.8.1.6 (*Emsley and Cowtan, 2004*) and refinement was carried out using PHENIX 1.12–2829 (*Adams et al., 2010*). Initial rigid body refinement resulted in R/$R_{free}$ values of 26.5/28.3%. Using the extended resolution improved the electron density maps and allowed placement of additional water molecules, two formate molecules (present at 3.5 M in the crystallization buffer), N-terminal residues 1–8, and an alternate chain path for residues 153–165, improving R/$R_{free}$ to 20.9/24.8%. Residues 180–181 are at the tip of a disordered loop and were not modeled, and residues 293–294 and the C-terminal His-tag beyond it are not visible in the electron density. Electron density is weak in several regions including the N-terminus and several loops, but the chain path was clear enough to build at least the backbone atoms for these residues. In late stages of refinement, TLS was implemented using one group per chain, dropping R/$R_{free}$ to 17.9/20.9%. B-factor weights were optimized in the final refinement step, yielding final R/$R_{free}$ of 17.2/20.4% for the final AimA model (*Table 2*).

## E.coli *and* Aeromonas *growth curves*

*E. coli* and *Aeromonas* cultures were grown overnight (37° C and 30° C, respectively) from a glycerol freezer stock in Luria Bertani (LB) broth to stationary phase. The cultures were back-diluted 1:100 the next morning in LB and loaded into a transparent 96-well plate. We used a total volume of 200 µL per well. 2,2'-dipyridyl (Sigma-Aldrich) titrations in DMSO were administered in 10 µL treatments for the *Aeromonas* growth curves. For the *E. coli* growth curves, AimA and mLCN titrations were administered in 20 µL treatments, with dipyridyl added to 300 µM in the LB 1:100 back-dilution. Controls for the *Aeromonas* and *E. coli* growth curves were treatments of DMSO or protein buffer (300 mM NaCl, 50 mM potassium phosphate pH 7, 5 mM BME), respectively. Growth (OD 600 nm) at 30° C was monitored using a FLUOStar Omega (BMG LABTECH) plate reader. Readings were taken every 1 hr for up to 36 hr. Replicate curves (three or four) were plotted with standard deviation using Prism (GraphPad).

## Statistical analysis

The appropriate sample size for experiments quantifying intestinal neutrophils and bacterial colonization level was estimated *a priori* using a power of 84% and a significance level of 0.05. From previously published data on intestinal neutrophil quantification (*Rolig et al., 2015*) and zebrafish gut bacterial mono-association colonization (*Rolig et al., 2015*; *Hill et al., 2016*) we estimated a effect size of 0.24 for neutrophil influx and 0.35 for bacterial colonization. These parameters suggested using an n of 40 and 23 in order to detect significant changes between treatment groups for either neutrophil influx or bacterial colonization, respectively. Each experiment described herein contains about 10–15 biological replicates (individual fish per treatment group). These experiments of 10–15 fish were repeated multiple times (technical replicates), resulting in pooled data sets ranging from 20 to 70 biological replicates. Grubbs statistical outlier test was applied to the data and data points that met the criteria for a statistical outlier were removed. In the figures, these data are presented as

box and whisker plots, which display the data median (line within the box), first and third quartiles (top and bottom of the box), and minimum and maximum values (whiskers). All data points that generate the box and whisker plots are presented as individual points within the plot. These pooled data were analyzed through the statistical software Prism Graphpad software. For experiments measuring a single variable with multiple treatment groups, a single factor ANOVA with post hoc means testing (Tukey) was utilized. A p-value of less than 0.05 was required to reject the null hypothesis that no difference existed between groups of data. For survival curves a Mantel-Cox test was used to determine significance.

## Acknowledgements

We thank Rose Sockol and the UO Zebrafish Facility staff for fish husbandry. We thank Joerg Graf for providing the *Aeromonas veronii* HM21 strains and S James Remington for advice on crystallography.

## Additional information

### Competing interests

Annah S Rolig, Karen Guillemin: Have two patent applications related to AimA pending in the United States, Europe, and Canada (US Patent App. 15/883,999 and 15/311,667). The other authors declare that no competing interests exist.

### Funding

| Funder | Grant reference number | Author |
| --- | --- | --- |
| National Institute of General Medical Sciences | P50GM098911 | Karen Guillemin |
| National Institute of Diabetes and Digestive and Kidney Diseases | F32DK098884 | Annah S Rolig |
| Eunice Kennedy Shriver National Institute of Child Health and Human Development | P01HD22486 | Karen Guillemin |

The funders had no role in study design, data collection and interpretation, or the decision to submit the work for publication. The content is solely the responsibility of the authors and does not necessarily represent the official views of the NIH.

### Author contributions

Annah S Rolig, Conceptualization, Formal analysis, Funding acquisition, Investigation, Visualization, Methodology, Writing—original draft; Emily Goers Sweeney, Formal analysis, Investigation, Visualization, Writing—review and editing, Purified AimA and AimB, Determined the crystal structure of AimA; Lila E Kaye, Michael D DeSantis, Investigation, Performed genetic and functional characterizations of the Aim proteins; Arden Perkins, Investigation, Made significant intellectual contributions to the analysis of crystallographic data; Allison V Banse, Investigation, Characterized the secreted proteins from the type 2 secretion mutant and complemented strain of Aeromonas; M Kristina Hamilton, Investigation, Writing—review and editing, Performed genetic and functional characterizations of the Aim proteins; Karen Guillemin, Conceptualization, Resources, Supervision, Funding acquisition, Project administration, Writing—review and editing

### Author ORCIDs

Annah S Rolig [iD] https://orcid.org/0000-0001-7080-4352
Emily Goers Sweeney [iD] http://orcid.org/0000-0002-5539-8875
Karen Guillemin [iD] https://orcid.org/0000-0001-6004-9955

## Ethics

Animal experimentation: All zebrafish experiments were performed following protocols approved by the University of Oregon Institutional Animal Care and Use Committee (#15-98) and following standard zebrafish protocols.

## Decision letter and Author response

Decision letter https://doi.org/10.7554/eLife.37172.033
Author response https://doi.org/10.7554/eLife.37172.034

## Additional files

### Supplementary files

• Supplementary file 1. Mass spectrophotometry results comparing cell-free supernatant from the complement of the type II secretion mutant *A. veronii* strain Hm21 (WT, ΔT2C) compared to the type II secretion mutant (mut, ΔT2). Mass-spectrophotometry was performed on the CFS from ΔT2C and ΔT2 determined which proteins were enriched in the ΔT2C compared to the ΔT2 strain. This table lists the top 22 proteins that were enriched by greater than 10 counts in the ΔT2C CFS.
DOI: https://doi.org/10.7554/eLife.37172.027

• Supplemental file 2. Structural homology hits to the N and C terminal domains of AimA. Structural homology searches performed against the N and C terminal domains of AimA separately revealed that both domains have similarity to proteins in the calycin superfamily, primarily avidins and lipocalins.
DOI: https://doi.org/10.7554/eLife.37172.028

• Transparent reporting form
DOI: https://doi.org/10.7554/eLife.37172.029

### Data availability

Diffraction data have been deposited in PDB under the accession code 6B7L.

The following dataset was generated:

| Author(s) | Year | Dataset title | Dataset URL | Database and Identifier |
|---|---|---|---|---|
| Rolig AS, Sweeney EG, Kaye LE, De-Santis MD, Perkins A, Banse AV, Hamilton MK, Guillemin K | 2018 | Aeromonas Immune Modulator A Structure | https://www.rcsb.org/structure/6B7L | RCSB Protein Data Bank, 6B7L |

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
