## [Decision Letter]

Thank you for submitting your article "A bacterial immunomodulatory protein with lipocalin-like domains facilitates host-bacteria mutualism" for consideration by *eLife*. Your article has been reviewed by three peer reviewers, and the evaluation has been overseen by a Guest Reviewing Editor and Wendy Garrett as the Senior Editor. The following individuals involved in review of your submission have agreed to reveal their identity: Vijay Kumar (Reviewer #1); Francois Leulier (Reviewer #3).

The reviewers have discussed the reviews with one another and the Reviewing Editor has drafted this decision to help you prepare a revised submission.

Summary:

The manuscript by Rolig et al. reports the exciting discovery of a novel bacterial protein, AimA, and its role in mediating host-bacteria mutualism between zebrafish and *Aeromonas*spp. Structural analysis on AimA reveals that it is comprised of two lipocalin-like domains that share homology with mouse lipocalin 2 (mLcn2). Similar to mLcn2, each of AimA's lipocalin-like domains can modulate neutrophil responses. However, the genetic loss of AimA in *Aeromonas* results in heightened neutrophil infiltration to the gut, which in turn reduces *Aeromonas* fitness and survival. Such a defect in fitness is not apparent when mutant *Aeromonas* is mono-associated with the immunocompromised *myd88^-/-^*zebrafish. On the contrary, these *myd88^-/-^*zebrafish displayed a striking increase in mortality, which correlates with an overgrowth of *Aeromonas* in the gut. Taken together, these intriguing observations shed new insight on the central role of AimA in mediating a distinct bi-directional interaction between a commensal bacterium and its host. Overall, the study is well-designed and supported using robust experimental approach and genetic models.

Essential revisions:

1) In a seminal study (Goetz et al., 2002), Roland Strong and colleagues previously heterologously expressed mLcn2 in *E. coli* which led to the discovery that mLcn2 can sequester and co-purify with a bacterial siderophore, namely enterobactin. Despite the homology between mLcn2 and AimA, it remains unclear whether AimA has the ability to bind siderophore. It is therefore important to clarify whether AimA can bind (and co-purify with) enterobactin and ensure that the AimA isolated by the authors is siderophore-free. Can the authors also please comment on whether heterologous expression of AimA impeded the growth of the *E. coli*?

2) To make this article more accessible to a diverse readership, many of whom may be unfamiliar with the nuances of the zebrafish model, please discuss the important limitations of the model (e.g. that it is highly artificial, examines association with a single species from a complex consortium that normally colonize zebrafish, limitations of the larval GF model, and the effect sizes of the inflammatory and viability phenotypes, which show rather subtle differences between experimental groups. E.g. in some cases, just 1-3 neutrophils per entire gut. These differences are likely not to be well-appreciated by those readers not intimately familiar with this zebrafish larval model.

3) Please justify the use of the timepoints in experiments. Why are some 6 days and others 7 days, and why were these particular timepoints chosen?

4) Please discuss the fate of the larvae infected with the various mutants outside of the measured time points. Would any of these GF or *Aeromonas*-associated larvae develop into adulthood? And if not, what other possible explanations could there be for rather subtle cellular differences in larvae that are not destined to survive into adulthood?

5) It is traditional to show the phenotype of the WT bacterial strain in any in vivo assay. However, the WT strain is not presented in Figure 1, on the complemented mutant. It is not always the case that a complemented mutant shows a WT phenotype, so this would be important to establish.

6) In the subsection “*Aeromonas* secretes a protein that modulates the intestinal neutrophil response”, it states that various mutant colonize to similar levels. However, it is latter concluded that the immunomodulatory proteins is important to support colonization of *Aeromonas* (subsection “AimA controls host neutrophil response and promotes *Aeromonas* colonization”). Please clarify how fitness and colonization are differentiated? Most would consider that colonization reflects fitness over the course of a short term assay like those used here.

---

## [Author Response]

Essential revisions:1) In a seminal study (Goetz et al., 2002), Roland Strong and colleagues previously heterologously expressed mLcn2 in E. coli which led to the discovery that mLcn2 can sequester and co-purify with a bacterial siderophore, namely enterobactin. Despite the homology between mLcn2 and AimA, it remains unclear whether AimA has the ability to bind siderophore. It is therefore important to clarify whether AimA can bind (and co-purify with) enterobactin and ensure that the AimA isolated by the authors is siderophore-free. Can the authors also please comment on whether heterologous expression of AimA impeded the growth of the E. coli?

We did not find enterobactin in the AimA protein crystal structure. We can confirm that the purified AimA is siderophore free because the *E. coli* strain, BL21 DE3, which we used to express AimA does not produce enterobactin; we now state this in the manuscript (subsection “Presence of mouse lipocalin-2 inhibits AimA function”, last paragraph). In response to the reviewers’ comment we now present the following evidence that AimA does not bind siderophores:

1) Careful structural examination of the wide enterobactin binding cleft in LCN2 and comparison of the analogous structural regions of each of the domains of AimA reveals that the openings of N- and C-term calyx domains of AimA are much narrower, are occluded by loops, and lack the polar enterobactin-interacting residues of LCN2. This detailed structural analysis is now presented in the last paragraph of the subsection “Presence of mouse lipocalin-2 inhibits AimA function” and in Figure 3—figure supplement 1A-F.

2) We tested whether AimA binds enterobactin by growing *E. coli* K12 under iron limiting conditions that make it dependent on the siderophore function of its secreted enterobactin. As expected under these conditions, addition of purified LCN2 inhibited *E. coli* growth. In contrast, addition of the same concentration range of purified AimA had no effect on *E. coli* growth. These data are now presented in the last paragraph of the subsection “Presence of mouse lipocalin-2 inhibits AimA function” and in Figure 3—figure supplement 1G.

3) We tested whether AimA functions in iron acquisition for *Aeromonas* by testing the ability of wild type *Aeromonas* and the Aer Δ*A*Δ*B* mutant to grow in iron-depleted media containing dipyridyl. We found that the two strains had identical growth curves under iron limitation. These data are now presented in the second paragraph of the subsection “AimA controls host neutrophil response and promotes *Aeromonas* colonization” and in Figure 5—figure supplement 1C, suggesting that AimA and AimB do not function in iron acquisition for *Aeromonas*.

Based on this evidence, we concluded that AimA does not bind enterobactin (Discussion, fourth paragraph).

2) To make this article more accessible to a diverse readership, many of whom may be unfamiliar with the nuances of the zebrafish model, please discuss the important limitations of the model (e.g. that it is highly artificial, examines association with a single species from a complex consortium that normally colonize zebrafish, limitations of the larval GF model, and the effect sizes of the inflammatory and viability phenotypes, which show rather subtle differences between experimental groups. E.g. in some cases, just 1-3 neutrophils per entire gut. These differences are likely not to be well-appreciated by those readers not intimately familiar with this zebrafish larval model.

We agree that is important to state the implications and limitations of our model. We have now included (in the third paragraph of the Introduction) a description of the zebrafish gnotobiotic system and a discussion of its limitations.

3) Please justify the use of the timepoints in experiments. Why are some 6 days and others 7 days, and why were these particular timepoints chosen?

In our initial characterization of the neutrophil response to *Aeromonas* and the type 2 secretion mutant and subsequent analysis of AimA activity (Figures 1, 3, and 4) we performed the assay at 6 dpf because we have a considerable amount of data collected on neutrophil populations of germ-free, conventional, and mono-associated zebrafish at this time point (Rolig et al., 2015, Rolig et al., 2017). When we examined the colonization dynamics of the *aim* mutants, we saw the greatest colonization defect at 7 dpf. We therefore subsequently analyzed the host neutrophil phenotypes in response to these strains at 7 dpf in Figures 5 and 6. We now explain this rationale in the text at lines 321 and 338 and we have included a new Figure 5—figure supplement 2 that shows the colonization dynamics of the ZF Aer wild type and *aimA* mutant strains at days 5, 6, and 7.

4) Please discuss the fate of the larvae infected with the various mutants outside of the measured time points. Would any of these GF or Aeromonas-associated larvae develop into adulthood? And if not, what other possible explanations could there be for rather subtle cellular differences in larvae that are not destined to survive into adulthood?

We realize we did not make it clear that we can only readily maintain zebrafish in a gnotobiotic state through their larval stage (up to about 8 dpf) while they are still provisioned with nutrients from their yolk. We do not have robust methods for feeding and maintaining adult zebrafish in a germ-free state. We have now included this in our text about the limitations of the zebrafish gnotobiotic system in the third paragraph of the Introduction.

5) It is traditional to show the phenotype of the WT bacterial strain in any in vivo assay. However, the WT strain is not presented in Figure 1, on the complemented mutant. It is not always the case that a complemented mutant shows a WT phenotype, so this would be important to establish.

The complemented mutant (ΔT2C) was previously shown to restore T2SS function (Graf, 2011), thus in these experiments we chose to compare the ΔT2 strain to the complemented strain, which allowed us to rule out the possibility that phenotypes associated with the ΔT2 strain were due to a polar or second site mutation introduced by the mutagenesis. We now explain this rationale in the subsection “*Aeromonas* secretes a protein that modulates the intestinal neutrophil response”.

Because the parent strain Hm21 of the two secretion system mutants is the same parent strain we used to make the *aim* deletion mutants, the colonization of the wild-type parent and the intestinal neutrophil response to it are presented in Figure 5C and 5E.

6) In the subsection “Aeromonas secretes a protein that modulates the intestinal neutrophil response”, it states that various mutant colonize to similar levels. However, it is latter concluded that the immunomodulatory proteins is important to support colonization of Aeromonas (subsection “AimA controls host neutrophil response and promotes Aeromonas colonization”). Please clarify how fitness and colonization are differentiated? Most would consider that colonization reflects fitness over the course of a short term assay like those used here.

It is correct that the Aer ΔT2did not exhibit a colonization defect (Figure 1—figure supplement 1) in comparison to the Aer ΔT2C(complemented strain), whereas theΔ*A*Δ*B* mutant (Hm21 strain) and the Δ*AimA* mutant (zebrafish *Aeromonas* isolate ZOR001) did exhibit a colonization defect in comparison to their respective wild-type parent strains (Figure 5). We now address this in the third paragraph of the subsection “AimA controls host neutrophil response and promotes *Aeromonas* colonization” and point out that this could be due to other phenotypic consequences of the altered secretion profile of the Aer ΔT2strain.

We have further clarified our use of the term ‘fitness’ throughout the manuscript. We have replaced the term fitness with “the capacity to survive the intestine” or to “colonization advantage” to clarify our meaning.